# Interactive network configuration maintains bacterioplankton community structure under elevated $CO_2$ in a eutrophic coastal mesocosm experiment

Xin Lin[†*1], Ruiping Huang[†1], Yan Li[1], Futian Li[1], Yaping Wu[1,2], David A. Hutchins[3], Minhan Dai[1], Kunshan Gao[*1]

**Institutions:**

[1] State Key Laboratory of Marine Environmental Science, College of Ocean & Earth Sciences, Xiamen University, Xiamen 361102, PR China.

[2] College of Oceanography, Hohai Uuniversity, No.1 Xikang road, Nanjing 210000, China.

[3] Department of Biological Sciences, University of Southern California, 3616 Trousdale Parkway, AHF 301, Los Angeles, CA 90089-0371, USA.

[†] These authors contributed equally to this work.

*Correspondence to*: Xin Lin (xinlinulm@xmu.edu.cn, TEL: +865922880171);

Kunshan Gao (ksgao@xmu.edu.cn, TEL: +865922187963)

**Abstract**
There is increasing concern about the effects of ocean acidification on marine biogeochemical and
ecological processes and the organisms that drive them, including marine bacteria. Here, we examine the
effects of elevated $CO_2$ on the bacterioplankton community during a mesocosm experiment using an
artificial phytoplankton community in subtropical, eutrophic coastal waters of Xiamen, Southern China.
Through sequencing the bacterial 16S rRNA gene V3-V4 region, we found that the bacterioplankton
community in this subtropical, high nutrient coastal environment was relatively resilient to changes in
seawater carbonate chemistry. Based on comparative ecological network analysis, we found that
elevated $CO_2$ hardly altered the network structure of high abundance bacterioplankton taxa, but appeared
to reassemble the community network of low abundance taxa. This led to relatively high resilience of the
whole bacterioplankton community to the elevated $CO_2$ level and associated chemical changes. We also
observed that the Flavobacteria group, which plays an important role in the microbial carbon pump,
showed higher relative abundance under the elevated $CO_2$ condition during the early stage of the
phytoplankton bloom in the mesocosms. Our results provide new insights into how elevated $CO_2$ may
influence bacterioplankton community structure.
**Key words:** elevated $CO_2$; mesocosm; bacterioplankton community; ecological network; Flavobacteria
**1 Introduction**
It is well established that ocean acidification is being caused by increased uptake of
anthropogenically-derived carbon dioxide in the surface ocean. Consequently, it is predicted that under a
"business-as-usual" $CO_2$ emission scenario, the present average surface pH value will drop 0.4 units over
the next century (Gattuso et al., 2015). Despite a growing interest in the importance of the roles of marine
bacterioplankton in ocean ecosystems and biogeochemical cycles, our current understanding of their
responses to ocean acidification is still limited. Over half of autotrophically-fixed oceanic $CO_2$ is
processed by heterotrophic bacteria and archaea through the microbial loop and carbon pump (Azam,
1998; Jiao et al., 2010). Furthermore, marine bacterioplankton play an essential role in marine
ecosystems and global biogeochemical cycles central to the biological chemistry of Earth (Falkowski et
al., 2008). The null hypothesis is that elevated $CO_2$ will not affect biogeochemical processes (Liu et al.,
2010; Joint et al., 2011), however more investigation is required. Ocean acidification mesocosm
experiments provide good opportunities to explore the responses of marine bacteria to elevated $CO_2$.
Mesocosm studies conducted in the Arctic Ocean, Norway, Sweden and the coastal Mediterranean Sea
using natural phytoplankton communities have often found that elevated $CO_2$ has little direct effect on
the bacterioplankton community (Zhang et al., 2013; Ray et al., 2012, Roy et al., 2013; Baltar et al.,
2015). In contrast, phytoplankton blooms induced by high $CO_2$ can sometimes have significant indirect
effects on heterotrophic microbes, thus altering bacterioplankton community structure (Allgaier et al.,
2008; Hutchins and Fu, 2017).
Although most mesocosm studies have showed that elevated $CO_2$ had an insignificant impact on
bacterioplankton community structure, microcosm experiments have demonstrated that small changes in
pH can have direct effects on marine bacterial community composition (Krause et al., 2012). Ocean
acidification experiments using natural biofilms showed bacterial community shifts, with decreasing
relative abundance of Alphaproteobacteria and increasing relative abundance of Flavobacteriales (Witt et
al., 2011). Coastal microbial biofilms grown at high $CO_2$ level also showed different community
structures compared to those grown at ambient $CO_2$ level in a natural carbon dioxide vent ecosystem
(Lidbury et al., 2012). Ocean acidification also affects the community structure of bacteria associated
with corals. It has been reported that the relative abundance of bacteria associated with diseased and
stressed corals increased under decreasing pH conditions (Meron et al., 2011). A very limited number of
studies focused on the effects of ocean acidification on isolated bacterial strains have also been
reported. Under lab conditions, growth of *Vibrio alginolyticus*, a species belonging to the class
Gammaproteobacteria, was suppressed at low $CO_2$ levels (Labare et al., 2010). In contrast, stimulation of
growth was observed for one Flavobacteria species under high $CO_2$ levels (Teira et al., 2012).
Taken together, results from mesocosm, microcosm and cultured isolate experiments indicate a
potentially complex interaction between different groups of marine bacteria in response to elevated $CO_2$.
One promising method to elucidate these types of complex interactions is network analysis. Ecological
network approaches have been successfully applied to investigate the complexity of interactions among
zooplankton and phytoplankton from different trophic levels during the Tara Oceans Expedition project
(Lima-mendez et al., 2015; Guidi et al., 2015). Elucidating the complex interactions between
bacterioplankton and other marine organisms under anthropogenic perturbations will increase our
understanding of their impact in a holistic way. Previous studies using ecological network analysis
showed that elevated $CO_2$ significantly impacted soil bacterial/archaeal community networks, by
decreasing the connections for dominant fungal species and reassembling unrelated fungal species in a
grassland ecosystem (Tu et al., 2015). It was also reported using ecological network analysis that
elevated $p\text{CO}_2$ did not significantly affect microbial community structure and succession in the Arctic
Ocean, suggesting bacterioplankton community resilience to elevated $p\text{CO}_2$ (Wang et al., 2016).

3       It has been reported that eutrophication problems in coastal regions lead to complex cross-linkages

between ocean acidification and eutrophication (Cai et al., 2011). The occurrence of ocean acidification
combined with other environmental stressors such as eutrophication can potentially produce synergistic
or antagonistic effects on bacterioplankton that differ from those caused by ocean acidification alone.
Although there are some reports from mesocosm experiments describing the response of bacteria to
elevated $\text{CO}_2$, there are limited studies on how the bacterial community responds to ocean acidification in
eutrophic marine environments. In this study, Illumina sequencing of the V3-V4 region of the bacterial
16S rRNA gene was used to explore the effects of ocean acidification on bacterioplankton community
composition and ecological network structure in a eutrophic coastal mesocosm experiment.
**2 Methods**
**2.1 Mesocosm setup and carbonate system manipulation**
The mesocosm experiment was conducted in the FOANIC-XMU (Facility for the Study of Ocean
Acidification Impacts of Xiamen University) mesocosm platform located in Wuyuan Bay, Xiamen,
Fujian province, East China Sea (N24°31′48″, E118°10′47″) during the months of December 2014 and
January 2015 (Fig. S1). Each transparent thermoplastic polyurethane (TPU) cylindrical mesocosm bag
was 3 m deep and 1.5 m wide (~4000 L total volume). After setting up the mesocosm bags within steel
frames, in situ seawater from Wuyuan Bay was filtered through a 0.01μm water purifying system and
used to simultaneously fill eight bags within 24 hours. The initial in situ seawater $p\text{CO}_2$ in Wuyuan Bay
was ~650 μatm, due to the active decomposition of land-sourced organic compounds. In order to reach
the target low $p\text{CO}_2$ associated with ambient air (400 ppm), $\text{Na}_2\text{CO}_3$ was added to each mesocosm to
increase dissolved inorganic carbon (DIC) and total alkalinity (TA) by 100 μmol/L and 200 μmol/L
respectively, based on carbonate system calculations (Lewis and Wallace, 1998). To adjust seawater to
projected end of this century seawater conditions of ~1000 ppm $CO_2$, about 5 L of $CO_2$ saturated filtered
seawater was added to 4 mesocosms (#2, #4, #7, #9), collectively considered to be the HC treatment,
while the other 4 mesocosms (#1, #3, #6, #8) were considered to be the LC treatment. Throughout the
experiment, HC mesocosms and LC mesocosms were bubbled with air containing 1000 ppm and 400
ppm $CO_2$, respectively supplied by $CO_2$ Enrichlors (CE-100B, Wuhan Ruihua Instrument & Equipment
Ltd, China) at a flow rate of 4.8 L per minute. Two diatoms, *Phaeodactylum tricornutum* CCMA 106
from the Centre for Collections of Marine Bacteria and Phytoplankton of the State Key Laboratory of
Marine Environmental Science (Xiamen University, China), and *Thalassiosira weissflogii* CCMP 102
from the Provasoli-Guillard National Center for Culture of Marine Phytoplankton (CCMP, USA), as well
as the coccolithophorid *Emiliania huxleyi* CS-369 from the Commonwealth Scientific and Industrial
Research Organization (CSIRO, Australia) were used as inoculum to construct a model phytoplankton
community. The effects of ocean acidification on these phytoplankton species mentioned above have
been intensively studied in the lab at physiological, biochemical and molecular levels. However, it is
difficult to extrapolate the response of these species to ocean acidification in natural complex
environments based on laboratory single species experiments (Busch et al., 2015). Our experiment was
designed as an intermediary step between laboratory and natural community field experiments, with
isolates of non-axenic phytoplankton being added to filtered natural waters. In this way, we were able to
investigate the effect of OA on phytoplankton and bacterioplankton in naturally eutrophic waters while
minimizing the complexity of shifting compositions of natural phytoplankton communities. Correlated
data about the effects of ocean acidification on the artificial phytoplankton community using the same
mesocosm system are available in (Jin et al., 2015) and (Liu et al., 2017).

2       The initial concentration of both *P. tricornutum* and *T. weissflogii* was 10 cells/mL, and *E. huxleyi* was

added at 20 cells/mL. The phytoplankton cultures were not axenic. The bacteria community
composition in the inoculated phytoplankton culture is shown in Fig. S2. Bacteria were not detectable
by flow cytometry in the filtered seawater just before inoculation. The three species of non-axenic
phytoplankton with bacterioplankton were mixed and then inoculated into each mesocosm bag. Thus, we
considered the initial bacterioplankton community to be the same or similar in each mesocosm bag
because the phytoplankton culture with bacterioplankton were evenly distributed into each mesocosm
bag for inoculation. The mesocosm and the $CO_2$ bubbling system were not sterile and not completely
closed during the experiment. Therefore, natural bacterioplankton were undoubtedly introduced into the
mesocosm system through aeration and air-sea exchange, and the bacterioplankton community in this
mesocosm experiment was derived from both the bacteria added with the inoculated phytoplankton
culture, and the natural local prokaryotic assemblage.

14       The use of the natural phytoplankton and bacterioplankton communities in this mesocosm experiment

would better represent the effects of ocean acidification on natural phytoplankton and bacterioplankton
communities. However, considering the highly eutrophic in situ seawater in Wuyuan Bay, it was
impractical to use the in situ seawater with the in situ natural community (bacterioplankton,
phytoplankton, zooplankton) directly without filtration, because of the dense phytoplankton bloom that
could be induced within several days, making the $p$CO$_2$ very difficult to keep under control. Alternatively,
we would have had to dilute 4 tons of seawater in the mesocosm bags at least every two days to maintain
the cell density and $CO_2$ concentration. Furthermore, considering a number of studies on the typical
phytoplankton responses to OA that have been carried out in laboratory, it was indeed a natural
progression for us to use typical model phytoplankton species to initiate the mesocosm studies before
using natural communities. Therefore, using the filtered seawater with inoculated isolates was reasonable
and logistically practical for our experiment.
**2.2 Bacteria sampling, filtration and sample selection**
A total of 500 mL to 2 L of water, depending on bacterial concentration, was collected from the
mesocosms. Six of the mesocosms (HC: #2, #4, #7 and LC: #1, #6, #8) were chosen for further study.
The inter-replicate variation in mesocosm experiments is usually more significant than in lab
experiments, because mesocosm experiments are conducted in open environments. Initially we had 4
replicates for each treatment, however, mesocosm bag 9 had a hole and mesocosm bag 3 was
contaminated by other phytoplankton in the beginning. Therefore, we did not consider the data from
these two compromised bags. Furthermore, three replicates of each treatment in our experiment to some
extent balanced out the bacteria introduction contingency, although the inter-replicate variation was
significant. Samples from days 4, 6, 8, 10, 13, 19, and 29 were collected in this study due to time,
personnel and equipment constraints. Sequential size fractionated filtration (2 μm and 0.2 μm
polycarbonate filters) by peristaltic pump was used to filter seawater collected from the mesocosm bags.
We tried to do sampling at day 2, but the samples were not successfully collected, probably due to very
high concentration of TEP (Transparent Exopolymer Particles) which easily blocked the polycarbonate
filter. Some replicates were missing at day 4 because we were able to successfully extract enough DNA
for sequencing only from bag 1, bag 7 and bag 6, also probably due to high TEP at day 4. It has been
reported that high TEP concentration was associated with high bacteria biomass (Sugimoto et al.,
2007, Ramaiah et al., 2000). According to the bacterioplankton abundance data (Fig. S3,Yibin Huang
et al.), the average bacterioplankton abundance was $6.69 \times 10^9$ cells/ml and $9.71 \times 10^9$ cells/ml at day 2
and day 4 respectively.
**2.3 DNA extraction, 16S rDNA V3-V4 region amplification and Illumina MiSeq sequencing**
Samples collected by 0.2 μm polycarbonate filters as described above were washed with PBS buffer and
then centrifuged at 9600g to obtain a cell pellet. A previously described DNA extraction protocol
(Francis et al., 2005) was utilized with some modifications, using the columns for DNA purification
from a bacteria DNA extraction kit (Tiangen DP302, China). Amplification, library construction and
sequencing were performed offsite at ANNOROAD using the DNA samples isolated as described above.
Primers were 341F (5'-CCTACGGGNGGCWGCAG-3') and 805R
(5'-GACTACHVGGGTATCTAATCC-3'), targeting the V3-V4 hyper variable regions of bacterial 16S
rRNA gene. The PCR amplification condition was as follows: initial denaturation at $95^{o}C$ for 3 min, 25
cycles of denaturation at $95^{o}C$ for 30 s, annealing at $55^{o}C$ for 30 s and extension at $72^{o}C$ for 30 s, then
final extension at $72^{o}C$ for 5 min. DNA library construction and sequencing followed the MiSeq Reagent
Kit Preparation Guide (Illumina, USA).
**2.4 Sequence assignment and sequence statistics analysis**
Clean paired-end reads were merged using PEAR (Zhang et al., 2014). The remaining raw sequences
were distinguished and sorted by unique sample tags. Unique operational taxonomic units (OTUs) were
picked against Greengenes database (http://greengenes.lbl.gov/cgi-bin/JD_Tutorial/nph-16S.cgi)
(McDonald et al., 2012) at 97% identity. OTUs with less than 2 reads were not considered. According to
the reference database, the representative sequences for each OTU were aligned using PyNAST
(Caporaso et al., 2010a). Finally, the phylogenetic tree was generated from the Graphlan (Langille
et al., 2013) using information on both the relative abundance and phylogenetic relationship of
observed species. QIIME 1.8.0 was used for sequence analysis including OTUs extraction for
bacterioplankton community structure analysis, OTUs overlapping analysis, species diversity, species
richness analysis and Principal Components Analysis (PCA) (Caporaso et al., 2010b). Bacterioplankton
community composition differences were assessed by Unweighted UniFrac distance using QIIME 1.8.0
as well. Dissimilarity tests were based on the Bray-Curtis dissimilarity index using analysis of
similarities (ANOSIM) (Clarke, 1993), non-parametric multivariate analysis of variance (ADONIS)
(Anderson, 2001), and multi-response permutation procedures (MRPP) (Mielke et al., 1981). Observed
species, Chao index, Shannon index and Simpson index were used for estimating the community
diversity. Analysis of variance (ANOVA) followed by T-test was performed to determine any significant
differences between HC and LC treatments.
**2.5 Ecological network construction and analysis**
As previously described, ecological network construction and analyses were performed based on the
relative abundance of OTUs in HC and LC treatments with three biological replicates
(http://129.15.40.240/mena/, Wang et al., 2016). The sequencing data from each mesocosm bag with
time series throughout the experiment were considered as different replicates. First, the similarity
matrices of the relative abundance of OTUs in LC and HC conditions were created respectively using
Pearson correlation coefficient across time points with biological replicates by a random matrix theory
(RMT)-based approach. Cut-off values were determined according to $R^2$ of power-law larger than 0.8
and equal between two manipulations to construct network structure. In order to ensure the constructed
networks were not random, biologically meaningless networks, 100 networks from the same matrix were
constructed and randomized. This resulted in the experimental networks being different from random
networks judging by significantly higher modularity, clustering coefficient and geodesic distance (Table
1). Then, module separation was produced using greedy modularity optimization, and *Z-P* values for all
nodes were calculated. In addition, to compare networks, the network connection was randomly rewired
and network topological properties were calculated. Finally, the bacteria network interaction was
visualized by Cytoscape v.3.3.0. The *Z–P* plots were constructed based on within-module (*Z*) and
among-module (*P*) values of each node derived from ecological network analysis. Ecological network
analysis is a novel RMT-based framework for studying microbial interactions. A node in ecological
network analysis shows an OTU and a link demonstrates a connection between two OTUs. The shortest
path between nodes is indicated by geodesic distance. Since the network constructed by OTUs can be
separated into several sub-communities, or modules, the modularity value indicates how well a network
can be divided into different sub-communities. Clustering coefficients demonstrate how well an OTU is
connected with other OTUs, while average clustering coefficients indicate the extent of connection in a
network.
**3 Results**
**3.1 Environmental parameters and experimental timeline**
The initial inorganic nitrogen, $PO_4^{3-}$, and $SiO_3^{2-}$ concentrations were 70–75 μmol/L, 2.5–2.6 μmol/L, and
38–39 μmol/L, respectively. Except for $SiO_3^{2-}$, nutrient concentrations decreased with rapid growth of
the phytoplankton and reached low concentrations by day 15. The dissolved total inorganic nitrogen
dropped from an initial concentration of 74.9 ± 2.87 μmol/L to 57.2 ± 4.37 μmol/L in the HC condition
and 72 ± 5.90 μmol/L to 53.6 ± 5.60 μmol/L in the LC condition by day 8, and reached low
concentrations by day 15 (average 3μmol/L in LC and average 6μmol/L in HC ).
$pH_{NBS}$ was determined on the scene with a pH/mV/ORP Meter (LEAN) calibrated with National
Bureau of Standards (NBS) buffers. Samples for DIC measurement were collected into 250 ml brown
borosilicate glass bottles and poisoned with 250 μL saturated $HgCl_2$ solution. DIC was determined by
acidification of 0.5 mL samples and subsequently infrared quantification of $CO_2$ with an Apollo® DIC
Analyzer. $pH_{total}$ was determined using a Orion 3 Star pH Benchtop analyzer and a Orion Ross
combined pH electrode, which was calibrated against three NIST-traceable pH buffers (pH 4.01, 7.00
and 10.01) (Cao et al. 2011). The $p$CO2 and TA values in this study were calculated from DIC and
$pH_{total}$ by the CO2SYS Program (Lewis and Wallace, 1998). The carbonate chemistry data at different
time points are shown in Table S1. A comprehensive description of carbonate chemistry measurements
and analysis during this mesocosm experiment is given in (Yan Li et al, unpublished). The initial $p$CO$_2$
of 373.0 ± 43.9 µatm ($pH_{NBS}$: 8.18 ± 0.02) in the LC treatment and 1296.0 ± 159.6 µatm ($pH_{NBS}$: 7.75 ±
0.04) in the HC treatment increased and reached a peak value of 922.5 ± 142.0 µatm ($pH_{NBS}$: 7.74 ± 0.08)
in the LC treatment at day 8 and 1879.6 ± 145.4 µatm ($pH_{NBS}$: 7.49 ± 0.05) in the HC treatment at day 4.
After reaching the peak, the $p$CO$_2$ values of both treatments decreased and were no longer statistically
different from day 13 onwards due to rapid $CO_2$ uptake by the phytoplankton, despite air containing 1000
ppm $CO_2$ being continuously bubbled into the HC treatments (Fig. 1 a, b). The bacterioplankton biomass
were very high on day 2 and day 4 (Fig. S3). However, the large amount of DIC (dissolved inorganic
carbon) produced by this high biomass of bacterioplankton could not be consumed by the phytoplankton
which were still at very low biomass, thus explaining the significant DIC production in the beginning.
The continuous rise of $p$CO$_2$ until the phytoplankton reached a certain concentration in the beginning was
also due to the high concentration of bacteria and the low concentration of phytoplankton, even though
the seawater was being aerated at target $p$CO$_2$. *P. tricornutum* and *T. weissflogii* were the dominant
species throughout the whole phytoplankton bloom in both HC and LC conditions. Chlorophyll *a* (Chl*a*)
concentration and diatom cell densities were used to identify changes in the diatom bloom following
inoculation (Fig. 1c, Liu et al., 2017). Chl*a* concentration increased from 0.23 ± 0.12 µg/L to 5.33 ± 1.82
μg/L in the LC conditions, and from 0.19 ± 0.07 μg/L to 5.75 ± 1.17 μg/L in the HC conditions from day
4 to day 9. Thereafter, Chl*a* concentration increased significantly and peaked at 109.9 ± 38.04 μg/L in the
LC treatment and 108.6 ± 46.07 μg/L in the HC treatment at day 15. Subsequently, Chl*a* concentrations
in both treatments were maintained at high concentrations until day 25 and decreased progressively
afterward. The bloom process identified by cell concentration of *P. tricornutum* and *T. weissflogii* was
similar with that illustrated by Chl*a* concentration. The growth of these two diatom species entered into
logarithmic phase from day 2. Cell density reached highest concentration at day 15 and day 19 for *T.*
*weissflogii* and *P. tricornutum* respectively, and then dropped down slowly. The coccolithophore
*Emiliania huxleyi* largely disappeared from the experimental mesocosms. A comprehensive description
of phytoplankton cell density, Chl*a* concentration, particle organic carbon (POC) and particle organic
nitrogen (PON) during the experiment is given in (Liu et al., 2017).
**3.2 Overview of sequencing analysis**
Following sequencing, 828524 high quality sequences were kept after processing (Table. S2), and 39.3%
of assembled reads were successfully aligned with the database. As a result, a total of unique 557
OTUs were generated after clustering at a 97% similarity level.  49.1% of OTUs were classified to
genera level with high taxonomic resolution (Table. S3). The phylogenetic tree was constructed based on
the sequences derived from all of the samples (Fig. S4). The bacterioplankton from all of the samples in
this study were identified as members of Bacteriodetes or Proteobacteria phylums. The most dominant
OTUs were Alphaproteobacteria, Rhodobacterales, Rhodobacterceae and Sediminicola at class, order,
family and genus level respectively (Fig. S5). The most abundant sequences at class, order, family and
genus levels accounted for 43.4 %, 42.6 %, 41.7% and 32.8 % of all sequences respectively.

**3.3 Bacterioplankton community structure throughout the phytoplankton bloom**

The bacterioplankton community structure in the mesocosm bags was very different from that in the originally inoculated phytoplankton cultures by day 4. For instance, some bacterioplankton phyla not detected in the original phytoplankton culture were observed in the samples collected on day 4. This may indicate that the bacterioplankton from the natural environment gradually became dominant in the mesocosm bags from day 0 to day 4. For example, Epsilonbacteria appeared in the mesocosms at day 4, while no Epsilonbacteria were detected in the coccolithophore or diatom cultures. Nearly 50% of the bacterioplankton in the mesocosms were composed of Epsilonbacteria in D4.1 (Fig. S2, Fig. 2).

Bacterioplankton community structure underwent dynamic changes during the diatom bloom in both the HC and LC treatments, varying significantly at different stages of the phytoplankton bloom (Fig. 2). At the phylum level, the bacterioplankton were dominated by Proteobacteria, while the relative abundance of Bacteroidetes was very low when nutrients were replete and diatom biomass was not high. However, Bacteroidetes increased dramatically as diatom biomass increased, and began to drop down after reaching a peak at day 10 (Fig. 2 and Fig. 3). In contrast, Proteobacteria began to increase after reaching their lowest concentration at day 10.

The Alphaproteobacteria, Flavobacteria, and Gammaproteobacteria classes with high abundance in all samples were selected for further analysis. The proportion of the Gammaproteobacteria class from the Proteobacteria phylum was very high at the beginning of the experiment ($50.2 \pm 13.8$ % in the HC treatment and $44.1 \pm 6.4$ % in the LC treatment at day 6) and decreased throughout the duration of the experiment. On the other hand, the Alphaproteobacteria class, also from the Proteobacteria phylum, decreased from initially high proportions ($46.9 \pm 13.2$ % in the HC treatment and $43.9 \pm 11.6$ % in the LC treatment) at day 6 to low proportions at day 10 ($27.2 \pm 2.8$ %) in the HC treatment, but remained almost

unchanged (44.6 ± 7.5 %) in the LC treatment and increased to 63.2 ± 27.3 % in the HC treatment and
60.8 ± 32.7 % in the LC treatment at day 29 (Fig. 2 and Fig. 3). The relative abundance of the
Flavobacteria class from the Bacteroidetes increased from the beginning and reached a peak at day 10
(52.2 ± 4.2 % in the HC treatment and 24.8 ± 16.9 % in the LC treatment), then dropped down until day
19 (19.9 ± 2.2 % in the HC treatment and 18.0 ± 15.4% in the LC treatment) (Fig. 2 and Fig. 3). The
proportional variation of the Flavobacteriales order and the Rhodobacterales order showed similar trends
with the Flavobacteria class and the Alphaproteobacteria class, respectively, as shown in Fig. 2 and Fig.

8 3.

**3.4 The effects of elevated $CO_2$ on bacterioplankton community structure**
Bacterial community structures of the HC and LC treatments were compared at different sampling
time-points (Fig 2), and a dissimilarity test based on ANOSIM, MRPP and ADONIS methods showed
that no statistically significant differences were observed (Table 2). PCA analysis also agreed with the
dissimilarity test (Fig. S8). The bacterioplankton community diversity in all samples was estimated by
observed species, Chao index, Shannon index and Simpson index. Rarefaction curves showed no
remarkable differences in community diversity between HC and LC, regardless of the time point (Fig.
S6). In general, bacterioplankton community diversity in both HC and LC treatments followed the same
trend, in that it peaked at day 10 and declined for the remainder of the experiment (Fig. S7).
Although the general trend of bacterioplankton community structure variation was similar in both the
HC and LC treatments as described above, some groups of bacterioplankton showed different responses
to elevated $CO_2$ at some time points. Notably, Bacteroidetes (predominantly Flavobacteria) had a higher
average proportion in the HC treatment (52.2 % of Bacteroidetes and 52.2 % of Flavobacteria) than in the
LC treatment (25.2% Bacteroidetes and 24.8% Flavobacteria) at the early stage of the diatom bloom at
day 10 ($p$=0.049 and 0.053 respectively). In contrast Proteobacteria, especially the Alphaproteobacteria,
were observed to have lower proportion in the HC treatment (47.8 % of Proteobacteria and 27.2% of
Alphaproteobacteria) than in the LC treatment (74.8 % of Proteobacteria and 44.6% of
Alphaproteobacteria) at day 10 ($p$=0.049 and 0.019 respectively, Fig. 3). At a higher taxonomic level,
Flavobacteriales demonstrated higher relative abundance in the HC treatment (52.2 %) compared to the
LC treatment (24.8 %) at day 10 ($p$=0.053), while for Rhodobacterales the inverse pattern was observed
($p$=0.020). Moreover, Flavobacteriaceae were observed to have a relatively higher ratio in the HC
treatment (50.3 %) compared to the LC treatment (24.0 %) at day 10 ($p$=0.053), whereas
Rhodobacteraceae demonstrated the opposite pattern ($p$=0.021, Fig. 3). It is notable that
Alteromonadales, belonging to the Gammaproteobacteria, had a higher ratio in the HC treatment
compared to the LC treatment at day 19 and day 29, although this was not statistically significant ($p$=0.24
and 0.34 at day 19 and 29 respectively).
**3.5 The effects of elevated $CO_2$ on bacterioplankton community interactions**
Both HC and LC networks were dominated by Alphaproteobacteria, Gammaproteobacteria and
Flavobacteria, suggesting their vital roles in maintaining stability of microbial ecosystems under both
HC and LC conditions. The observation of more negative links compared to positive links indicates the
dominant relationship among bacterioplankton is competitive rather than mutualistic under both the HC
and LC treatments. The average connectivity and clustering coefficient of the network were higher in the
HC treatment than in the LC treatment, while geodesic distance and modularity value was higher in the
the LC treatment. Bacterioplankton formed more modules under the LC treatment, but were densely
connected in less modules under the HC treatment (Table 1, Fig. 4). However, as shown in Fig. 4, the
links among the OTUs with high abundance, 558885 (Rhodobacteraceae), 572670 (Rhodobacteraceae),
190052 (Flavobacteriaceae), 107130 (Flavobacteriaceae) and 4331023 (Rhodobacteraceae), were
positive in both HC and LC.
Interestingly, some nodes that were sparsely distributed in independent modules in the LC network
formed dense modules with high connectivity in the HC network (Fig. 4). As the OTUs connected within
a module, they could be considered as a putative bacterioplankton ecological niche (Zhou et al., 2010). It
is plausible that elevated $CO_2$ disrupted the connection between different bacterioplankton community
niches, but enhanced alternative connections among species within certain ecological niches. Within
module connectivity ($Zi$) and among-module connectivity ($Pi$) indexes were used to identify key module
members (Olesen et al., 2007, Fig. 5). In an ecological context, the peripherals may represent specialists,
while module hubs and connectors may be considered more as intra-module and inter-module generalists
respectively. Network hubs are usually considered as super-generalists (Deng et al., 2012). It is
interesting that the numbers of connectors that are considered as generalists were reduced, whereas
module hubs were increased under the HC treatment. However, two network hubs, the super-generalists
that are more important than module hubs and connecters, were detected in the LC network but not in the
HC network (Fig. 5).
**4 Discussion**
This study was designed to bridge the gap between lab cultures and field studies, with isolates of
non-axenic phytoplankton being added to filtered natural waters. The lab conditions possibly have
selected for a fast-growing bacterial community adapted to live with semi-continuous phytoplankton
culture. Therefore, the inoculated bacterioplankton were likely preconditioned to lab conditions in
semi-continuous phytoplankton cultures prior to the experiment. However, the bacterioplankton from
the natural environment gradually became dominant in the mesocosm bags from day 0 to day 4, based

on the comparison of the community at day 4 and the original community in the phytoplankton cultures. For instance, during these 4 days members of the Arcobacter genus (OTU 553961) and Pseudomonadaceae family (OUT 543958) were introduced from surrounding seawater into the mesocosm bags. The bacterial growth rates under eutrophic conditions are much higher than under oligotrophic conditions (White et al., 1991; Kirchman, 2016). The bacterial growth rate reached 16.2 $day^{-1}$ (0.675 $h^{-1}$) during a diatom bloom in a mesocosm experiment using seawater from Santa Barbara Channel amended with nutrients (Smith et al., 1995). Under simulated eutrophication conditions, the growth rates of bacteria from the Mediterranean sea ranged from 0.245 $h^{-1}$ to 0.853 $h^{-1}$ based on the data measured roughly every 24 hours in batch mesocosms (Lebaron et al., 1999). We would like to point out that our experiments were conducted in eutrophic coastal seawaters with reduced predatory grazing pressure due to seawater filtration, which could stimulate the net bacterial growth rate. In addition, some species belong to Pseudomonas group, one of the most abundant bacterioplankton group from outside, were reported to have high growth rates (Adav, 2008; Eagon, 1962). Therefore, we think choosing 0.5 $h^{-1}$ as the bacterial growth rate of the bacterioplankton is tenable. In our study, assuming the bacterioplankton concentration at day 2 representing the concentration of Pseudomonadaceae, one of the most abundant bacterioplankton groups from surrounding seawater, the concentration of Pseudomonadaceae at day 0 could be estimated based on the growth rate of 0.5 $h^{-1}$ and the bacterioplankton concentration ($6.693 \times 10^9$ cells/ml) at day 2. The estimated concentration of Pseudomonadaceae at day 0 was about 3 cells/ml. Therefore, the ratio of bacteria being continuously introduced to actual standing stocks in the mesocosms was low, which allowed us to detect potential $CO_2$ effects in this mesocosm experiment.

The seawater used in this mesocosm experiment was filtered natural seawater (through 0.01 μm filter)

in Wuyuan bay. Although no bacteria or phytoplankton were detected in the filtered seawater by flow
cytometry, high concentrations of DOM (dissolved organic matter) and other nutrients in the seawater
could not be filtered out. According to Yan Li et al (unpublished), the dissolved organic carbon (DOC)
concentration was 258.9 μmol/L in average at day 2. It was not surprising that bacterioplankton were
able to grow very quickly with such high concentrations of DOC. Because the
phytoplankton-associated bacterioplankton were presumably adapted to the phytoplankton cultures,
they were used to living in the artificial seawater, not the local seawater in Wuyuan Bay. As the local
bacterioplankton were presumably well adapted to local conditions (such as high DOC concentration)
in Wuyuan Bay, it is perhaps not surprising that they could easily outcompete the phytoplankton
culture-derived bacterioplankton. Although bacterioplankton from the phytoplankton cultures were
inoculated into the mesocosm system at the beginning of the experiment, they were mostly replaced by
the natural bacterioplankton community within several days. Therefore, the natural bacterioplankton,
not the original bacterioplankton from the phytoplankton culture, mainly determined the final responses
of the community to different $CO_2$ concentrations.

15        In this mesocosm experiment, significant variation in community structure was observed through the

whole diatom bloom process, suggesting that the diatom bloom was a major driver for bacterioplankton
community structure dynamics in both the HC and LC treatments. This finding is in line with previous
mesocosm experiments and field observations (Allgaier et al., 2008, Teeling et al., 2012). Along with
the phytoplankton bloom process, the inter-replicate variation of bacterioplankton community became
more apparent, which was inevitable for an outdoor mesocosm experiment. For example, the
bacterioplankton community in mesocosm bag 8 was dominated by *Phaeobacter. sp* at day 29, which
was distinct from the other mesocosm bags. According to the phytoplankton data mesocosm bag 8 was
probably contaminated with dinoflagellates at a late stage of the algal bloom, likely resulting in a
different bacterioplankton community structure compared to the others. In general, no statistically
significant differences were detected in this study, probably due to high variability among replicates. At
day 10 the inter-replicate-variability in the relative abundance of some groups of bacterioplankton was
relatively low, especially for the HC treatment. Indeed, statistically significant differences between the
HC and LC treatments in the abundances of certain groups of bacterioplankton were detected at day 10.
Therefore, only when the variability among replicates was smaller than the variability between
different treatments could statistically differences between treatments be detected.
Although effects of elevated $CO_2$ on bacterioplankton communities have been reported (Allgaier et al.,
2008; Tanaka et al., 2008; Wang et al., 2016; Zhang et al., 2013; Ray et al., 2012; Roy et al., 2013;
Baltar et al., 2015; reviewed in Hutchins and Fu, 2017), how marine bacteria communities react to the
occurrence of elevated $CO_2$ in eutrophic seawater is still uncertain. This mesocosm study
comprehensively investigated the effects of elevated $CO_2$ on bacterioplankton community structure and
networks using Illumina sequencing and ecological network analysis in the context of eutrophication.
Compared to the effects of the phytoplankton bloom, ocean acidification did not strongly influence the
bacterioplankton community structure. The results indicate that bacterial abundance and community
structure at different taxonomic levels were generally similar between the HC and LC treatments at the
different diatom bloom stages, in line with previous ocean acidification mesocosm bacterioplankton
community studies (Tanaka et al., 2008; Wang et al., 2016; Zhang et al., 2013; Ray et al., 2012; Roy et
al., 2013; Baltar et al., 2015). Differences in bacterioplankton community diversity between the HC and
LC treatments were also not remarkable. These results suggest the possibility that the whole
bacterioplankton community has a certain degree of resilience to elevated $CO_2$, which is consistent with
a previous stated hypothesis (Joint et al., 2011).
It has previously been proposed that the observed insignificant effects of ocean acidification on coastal
bacterioplankton may be due to their adaptation to strong natural variability in pH in coastal ecosystems,
where amplitudes of >0.3 units from diel fluctuations and seasonal dynamics are commonly seen
(Hofmann et al., 2011). The comparative ecological network analysis in this study to some extent
explains the resilience of the bacterioplankton community to elevated $CO_2$ levels. According to the
present study, substantial numbers of OTUs that were sparsely distributed in different and small modules
in the LC network became connected with each other and formed fewer modules in the HC network,
implying elevated $CO_2$ has the potential to reassemble the bacterioplankton community (Fig. 4). The
positive relationship among these principal components were almost unaltered in the network analysis,
suggesting that the positive relationships among them were robust in the face of $CO_2$ changes, thus
contributing to whole community stability (Fig. 4). It has also been reported that sparsely distributed
fungal species were reassembled into highly connected dense modules under long-term elevated $CO_2$
conditions (Tu et al., 2015).
It is noteworthy that the OTUs involved in possible community reassembly were not very abundant,
whereas the relationship between the abundant OTUs was virtually unaltered by elevated $CO_2$ in this
study. Although elevated $CO_2$ promoted the reassembly of the bacterioplankton community, the network
constructed by abundant OTUs which are usually considered as the foundation of the whole
bacterioplankton community was still stable in response to elevated $CO_2$. This to some extent led to
maintenance of bacterioplankton community structure under the ocean acidification stimuli in the
context of eutrophic conditions. Additionally, these data indicate that more negative than positive
relationships between OTUs were observed in both HC and LC treatments, which is consistent with a

previous ocean acidification mesocosm study conducted in the Arctic Ocean (Wang et al., 2016). It was

proposed that a community with more competitors would be more stable and yield less variation under

environmental fluctuations (Gonzalez and Loreau, 2009). Therefore, it could be speculated that the

dominant competitive relationship between bacterioplankton species in this mesocosm experiment

helped the whole bacterioplankton community to adapt to pH perturbations, with less variation in total

biomass and diversity.

Although the effects of elevated $CO_2$ on bacterioplankton community structure were not significant,

the proportion of some groups of bacterioplankton varied between the HC and LC treatments in the early

stages of the diatom bloom. Elevated $CO_2$ significantly increased the proportion of Flavobacteria

(dominated by Flavobacteriales) in the HC treatment at day 10, when the diatoms cells began to grow

rapidly. In contrast, the HC treatment had negative effects on the growth of Alphaproteobacteria

compared to the LC treatment. The results reported here are in line with previous reports about the

response of Flavobacteria to ocean acidification in biofilm and single species experiments (Witt et al.,

2011; Teira et al., 2012). Flavobacteria are considered as the "first responders" to phytoplankton blooms

because they specialize in attacking algal cells and further degrading biopolymers and organic matter

derived from algal detrital particles (Kirchman, 2002; Teeling et al., 2012). Flavobacteria are especially

good at converting high molecular weight (HMW) dissolved organic matter (DOM) to low molecular

weight (LMW) DOM using the highly efficient, extracellular, multi-protein complex TonB-dependent

transporter (TBDT) system, based on previous in situ proteomics and metatranscriptomics data (Teeling

et al., 2012). Higher abundance of Flavobacteria under elevated $CO_2$ means more HMW DOM could be

degraded and so enter into the carbon cycle (Buchan et al., 2014). Based on the results reported here, it

can be speculated that increased amounts of Flavobacteria under the elevated $CO_2$ treatment in eutrophic

seawater could promote the TBDT system to break down HMW DOM and lead to improved efficiency
of the Microbial Carbon Pump (MCP), and possibly further influence the carbon storage in the ocean
(Jiao et al., 2010). It has also been postulated that the Flavobacteria-originated, light-driven proton pump
proteorhodopsin could be involved in dealing with ocean acidification and pH perturbation (Fuhrman et
al., 2008). Recent metatranscriptomic data further emphasize the role of proteorhodopsin in pH
homeostasis in bacterioplankton under elevated $CO_2$ (Bunse et al., 2016; Gómez-Consarnau et al., 2007).
The underlying mechanisms underlying the enhanced growth of Flavobacteria under elevated $CO_2$ need
further investigation in the future.
Interestingly, Flavobacteria in our study showed higher abundance in the HC treatment in the early
phytoplankton bloom stage. However, a negative relationship between $CO_2$ level and relative abundance
of Bacteroidetes based on terminal restriction fragment length polymorphism (T-RFLP) method was
observed in a mesocosm experiment conducted in the Arctic region with low nutrient levels (Roy et al.,
2013). Moreover, the effects of elevated $CO_2$ on bacterioplankton community interaction webs in this
study were not observed in previous mesocosm work in the Arctic Ocean (Wang et al., 2016; Roy et al.,
2013). The results of the current study showed that the effects of elevated $CO_2$ in the context of
eutrophication were different compared to elevated $CO_2$ on bacterioplankton community networks in a
mesocosm study carried out in the oligotrophic Arctic Ocean. The data here and previously reported,
seemingly contradictory results highlight the importance of including the combined effects of ocean
acidification and other anthropogenic perturbations to interpret and predict the impact of global change
on marine life.
In this study, the majority of the particle-attached and algae-attached bacteria were filtered out by
sequential filtering. Additionally, the archaea were not included in our data because we used the
primers 341F/805R, which do not target archaea. Therefore, the community structure of
particle-associated bacteria and all archaea were not investigated in our study. Furthermore, a
simplified model phytoplankton community was used in this study, composed of the two diatom species
*P. tricornutum* and *T. weissflogii* in both LC and HC treatments. It is possible that the similarity of the
two bacterial communities in the two treatments was due to the similar composition and quality of DOM
produced by these two diatoms. With a more diverse natural phytoplankton community experimental
system, perhaps different phytoplankton taxa would have dominated in the HC and LC treatments,
leading to different bacterial communities. In future studies, it would also be worthwhile to sample over
a diel cycle in order to understand the cyclic variability in pH, and whether this affects short term changes
in bacterioplankton community structure.
**Conclusion**
Elevated $CO_2$ was not a strong influence on bacterioplankton community structure compared to the
diatom bloom process, based on 16S V3-V4 region Illumina sequencing. Based on ecological network
analysis, elevated $CO_2$ appeared to reassemble the community network of taxa present with low
abundance, but barely altered the network structure of the bacterioplankton taxa present with high
abundance. It is this differential sensitivity of common and rare groups to carbonate chemistry changes
that may largely explain the resilience of the bacterioplankton community to elevated $CO_2$.
**Author contributions**
Conceived and designed the experiments: K. Gao, X. Lin, M. Dai. Performed the experiments: R. Huang,
X. Lin, Y. Wu, Y. Li and F. Li. Analysed data: R. Huang and X. Lin. Wrote the paper: X. Lin. Revised
the paper: D. Hutchins and K. Gao. All authors reviewed the manuscript.
**Acknowledgments**
This study was supported by the National Key Research and Development Program of China (Grant No.
2016YFA0601302), the National Natural Science Foundation of China (No. 41306096 to X. Lin, No.
41430967 and No. 41120164007 to K. Gao), State Oceanic Administration of China
(SOA,GASI-03-01-02-04), The Open Fund of Key Laboratory of Marine Ecology and Environmental
Sciences, Institute of Oceanology, Chinese Academy of Sciences, and Laboratory of Marine Ecology
and Environmental Science, Qingdao National Laboratory for Marine Science and Technology
(KLMEES201608), Joint project of NSFC and Shandong province (Grant No. U1406403), Strategic
Priority Research Program of Chinese Academy of Sciences (Grant No. XDA11020302). DAH's
contributions were supported by U.S. NSF OCE 1260490 and 1538525, and his visits to Xiamen were
supported by "111" project from the Ministry of Education. We thank X. Liu, T. Xing, X. Cai, N. Liu, S.
Tong, X. Yi, T. Wang, H. Miao, Z. Li, D. Yan, W. Zhao and X. Zeng for their kind assistance in
operations of the mesocosm experiment.
**Competing interests:**
The authors declare no competing financial interests.

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

Availability of phosphate for phytoplankton and bacteria and of labile organic carbon for bacteria at
different pCO2 levels in a mesocosm study, Biogeosciences, (5), 669–678,
doi:10.5194/bgd-4-3937-2007, 2007.
Teeling, H., Fuchs, B. M., Becher, D., Klockow, C., Gardebrecht, A., Bennke, C. M., Kassabgy, M.,
Huang, S., Mann, A. J., Waldmann, J., Weber, M., Klindworth, A., Otto, A., Lange, J., Bernhardt, J.,
Reinsch, C., Hecker, M., Peplies, J., Bockelmann, F. D., Callies, U., Gerdts, G., Wichels, A., Wiltshire,
K. H., Glockner, F. O., Schweder, T. and Amann, R.: Substrate-Controlled Succession of Marine
Bacterioplankton Populations Induced by a Phytoplankton Bloom, Science (80-. )., 336(6081), 608–
611, doi:10.1126/science.1218344, 2012.
Teira, E., Fernández, A., Álvarez-Salgado, X. A., García-Martín, E. E., Serret, P. and Sobrino, C.:
Response of two marine bacterial isolates to high CO 2 concentration, Mar. Ecol. Prog. Ser., 453, 27–
36, doi:10.3354/meps09644, 2012.
Tu, Q., Yuan, M., He, Z., Deng, Y., Xue, K., Wu, L., Hobbie, S. E., Reich, P. B. and Zhou, J.: Fungal
Communities Respond to Long-Term CO 2 Elevation by Community Reassembly, Appl. Environ.
Microbiol., 81(7), 2445–2454, doi:10.1128/AEM.04040-14, 2015.
Wang, Y., Zhang, R., Zheng, Q., Deng, Y., Van Nostrand, J. D., Zhou, J. and Jiao, N.:
Bacterioplantkon community resilience to ocean acidification: evidence from microbial network
analysis, ICES J. Mar. Sci., 73(3), 865–875, doi:10.1093/icesjms/fst176, 2016.
White P.A., Kalff J., Rasmussen J. B. and Gasol J. M.: The Effect of Temperature and Algal Biomass
on Bacterial Production and Specific Growth Rate in Freshwater and Marine, Microb. Ecol., 21(2), 99–

14   118, 1991.

Witt, V., Wild, C., Anthony, K. R. N., Diaz-Pulido, G. and Uthicke, S.: Effects of ocean acidification
on microbial community composition of, and oxygen fluxes through, biofilms from the Great Barrier
Reef, Environ. Microbiol., 13(11), 2976–2989, doi:10.1111/j.1462-2920.2011.02571.x, 2011.
Worden, A. Z., Follows, M. J., Giovannoni, S. J., Wilken, S., Zimmerman, A. E. and Keeling, P. J.:
Rethinking the marine carbon cycle: Factoring in the multifarious lifestyles of microbes, Science
(80-. )., 347(6223), 1257594–1257594, doi:10.1126/science.1257594, 2015.
Zhang, J., Kobert, K., Flouri, T. and Stamatakis, A.: PEAR: a fast and accurate Illumina Paired-End
reAd mergeR, Bioinformatics, 30(5), 614–620, doi:10.1093/bioinformatics/btt593, 2014.
Zhang, R., Xia, X., Lau, S. C. K., Motegi, C., Weinbauer, M. G. and Jiao, N.: Response of
bacterioplankton community structure to an artificial gradient of pCO2 in the Arctic Ocean,
Biogeosciences, 10(6), 3679–3689, doi:10.5194/bg-10-3679-2013, 2013.
Zhou, J., Deng, Y., Luo, F., He, Z., Tu, Q. and Zhi, X.: Functional molecular ecological networks,
MBio, 1(4), e00169-10, doi:10.1128/mBio.00169-10.Editor, 2010.

## Figure legends

**Figure 1** Temporal variations of $p$CO$_2$ (a), pH$_{NBS}$ (b) and Chl$a$ (c) during the whole experiment. The

$p$CO$_2$ was calculated from DIC and pH using the CO2SYS program. Data are the means ± SD, n=3.

**Figure 2** Bacterioplankton community structure overview at different taxonomic levels during days 4, 6,

8, 10, 13, 19 and 29 (#1, #6, #8) under LC and HC (#2, #4, #7). X-axis represents sample name (for

example, D4.1 refers to bacterioplankton in mesocosm bag 1 collected at day 4) and the Y-axis

represents relative abundance of different groups of bacterioplankton.

**Figure 3** The relative abundance over time of primary taxa of the bacterioplankton community; HC in

red and LC in black. Proteobacteria (a) and Bacteroidetes (b) are phylum level; Flavobacteria (c) and

Alphabacteria (d) are class level; Flavobacteriales (e) and Rhodobacteriales (f) are order level;

Flavobacteriaceae (g) and Rhodobacteraceae (h) are family level. Data are the means ± SD (n=3), and the

asterisk represents a difference at $p<$ 0.05.

**Figure 4** Bacterioplankton network interactions under LC (a) and HC (b) conditions. Each node

represents an OTU. Node colors demonstrate different taxon. Each line connects two OTUs. A blue line

indicates a negative interaction between nodes, suggesting predation or competition, while a red line

indicates a positive interaction suggesting mutualism or cooperation. OTUs with importance are marked

with OTU identification numbers.
**Figure 5** Sub-modules in ecological network analysis under LC (a) and HC (b) conditions. Each dot
represents an OTU. The *Z–P* plot shows OTU distribution based on their module-based topological role
according to within-module (*Z*) and among-module (*P*) connectivity. The nodes were defined as module
hubs with *Zi* > 2.5 and *Pi* < 0.625, which were more closely connected within the module, while the
connectors were nodes with *Zi* < 2.5 and *Pi*> 0.625 were more closely connected to nodes in other
modules. Network hubs are super-generalist with a *Zi* >2.5 and *Pi* >0.625. The other nodes were
considered peripheral.

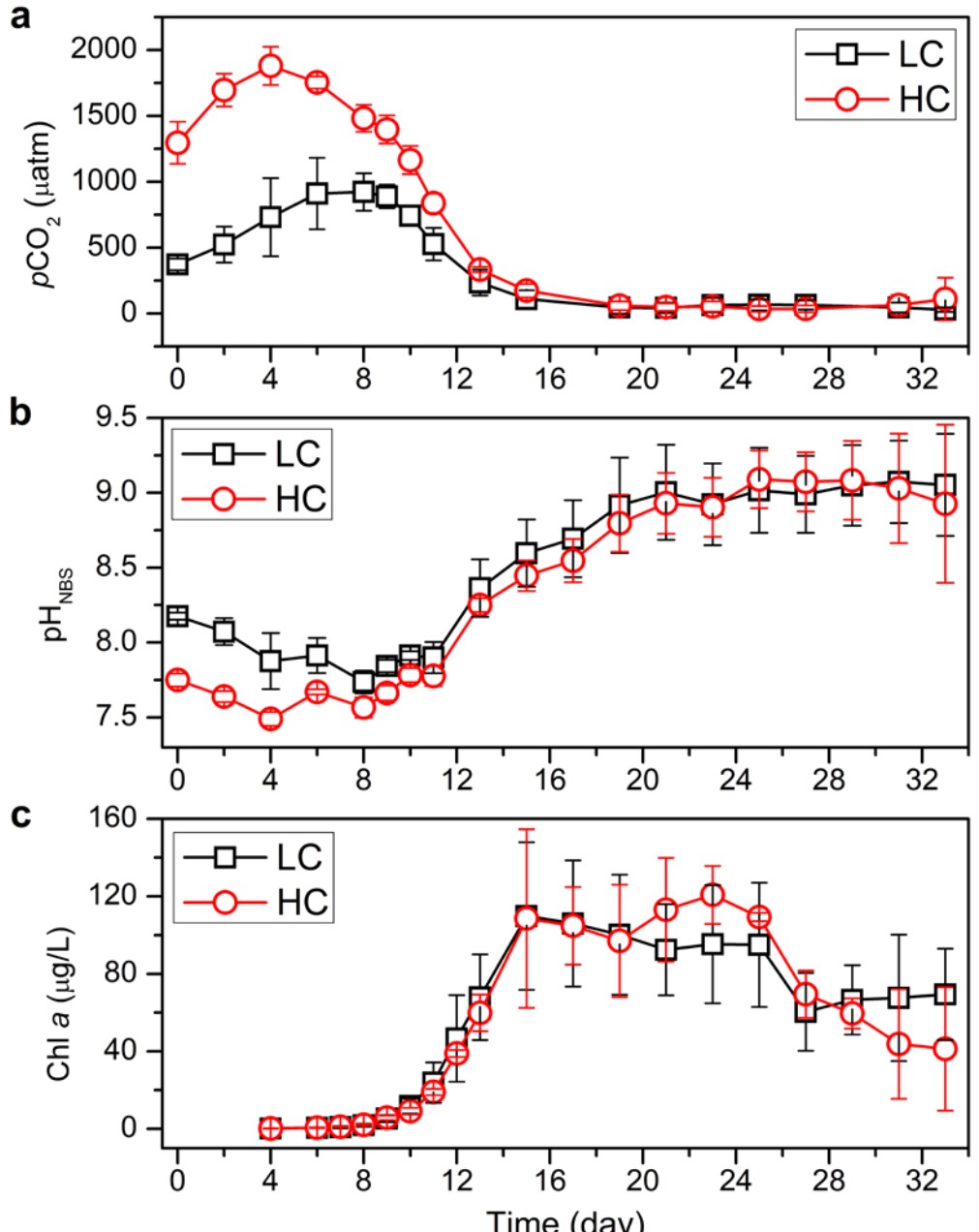

2                                    **Figure 1**

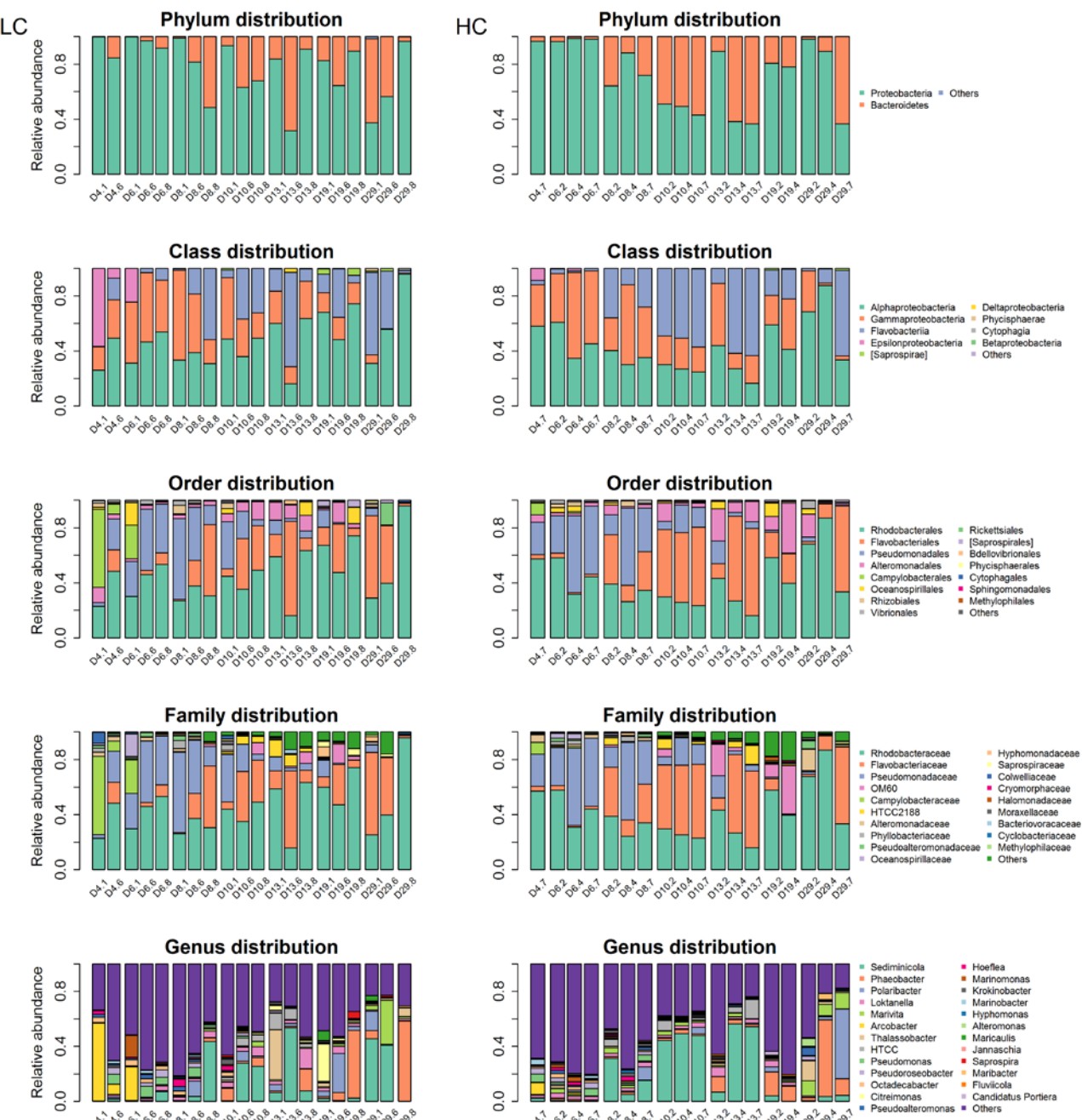

2 **Figure 2**

**Figure 3**

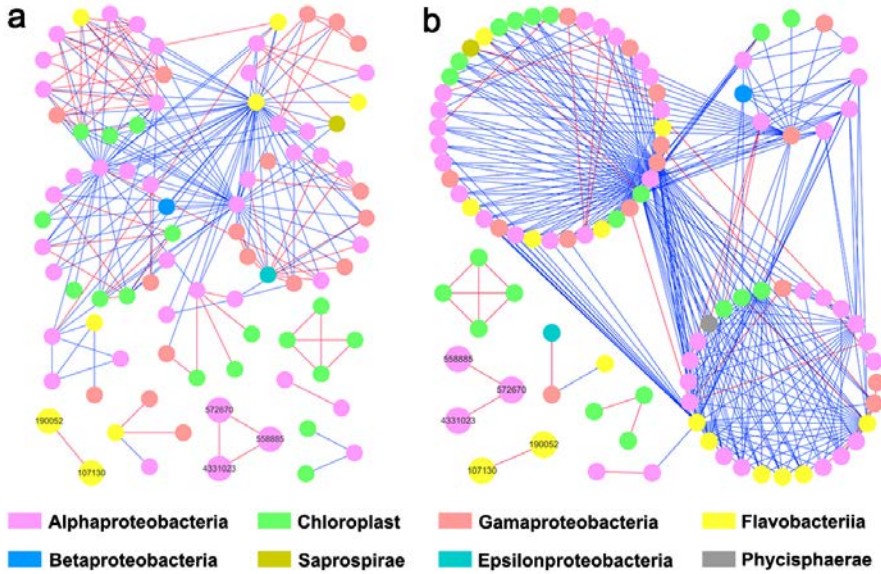

**Figure 4**

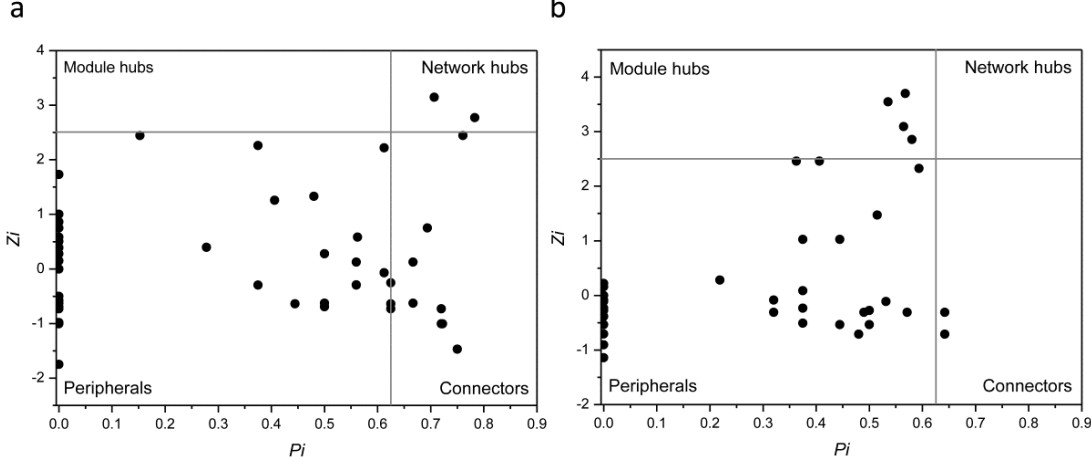

**Figure 5**

**Table 1** Topological properties of the bacterioplankton communities as represented by molecular networks under HC and LC treatments; also their rewired random networks.

| | Experimental network | | | | | | | Random network | | |
|---|---|---|---|---|---|---|---|---|---|---|
| | Total nodes | Total links | R2 of power-law | Average clustering coefficient (avgCC) | Average connectivity | Harmonic geodesic distance (HD) | Modularity | Average clustering coefficient (avgCC) | Harmonic geodesic distance (HD) | Modularity |
| LC | 85 | 209 | 0.817 | 0.402 | 0.625 | 3.397 | 0.414 | 0.424 +/- 0.023 | 2.187 +/- 0.049 | 0.249 +/- 0.010 |
| HC | 96 | 310 | 0.817 | 0.448 | 0.714 | 2.956 | 0.303 | 0.292 +/- 0.023 | 2.306 +/- 0.059 | 0.323 +/- 0.008 |

**Table 2** Dissimilarity tests of bacterial communities in the HC and LC treatments at various time points.

| | Anosim | | MRPP | | Adonis | |
|---|---|---|---|---|---|---|
| Time | R | P-value | δ | P-value | $R^2$ | P |
| day6 | -0.111 | 0.602 | 0.3952 | 1 | 0.15447 | 1 |
| day8 | 0.111 | 0.284 | 0.438 | 0.6 | 0.2 | 0.5 |
| day10 | 0.037 | 0.613 | 0.4929 | 0.7 | 0.17829 | 0.7 |
| day13 | 0.111 | 0.309 | 0.412 | 0.5 | 0.19714 | 0.5 |
| day19 | 0 | 0.693 | 0.4336 | 0.3 | 0.28263 | 0.3 |
| day29 | -0.259 | 1 | 0.4513 | 0.9 | 0.15517 | 0.9 |