# Peer review of "Interactive network configuration maintains bacterioplankton community structure under elevated $CO_2$ in a eutrophic coastal mesocosm experiment"

_Biogeosciences, 2017_

## Referee Comment (RC1) · Anonymous Referee #1 · 21 Feb 2017

The manuscript addresses the research question if bacterial communities in eutrophic coastal areas will be affected by elevated CO2 concentration. The topic is highly relevant given the possible effects of changes in oceanic carbon chemistry on bacterioplankton communities and subsequent biogeochemical nutrient cycles. The authors state that they found "insignificant effects of elevated CO2 on bacterioplankton communit(ies)", however their methodology and experimental setup is poor and insufficient to test the hypothesis.

The major criticisms of the manuscript is that the bacterial community composition (BCC) resulted from contamination of tubing and material used, as well as non-axenic phytoplankton cultures and hardly represents a natural bacterioplankton community.

[Figure]

Even if the bacteria found in the mesocosms were of marine origin, the initial community composition is unknown and not shown to be similar among the mesocosms. Therefore the results and study are not reproducible.

In fact, samples of the initial days are missing. The BCC after 4 days looks different between mesocosms, yet 3 replicates are missing in the figures, results section and statistical analysis without mentioning. Generally, it appears bizarre that a study addressing the BCC response to elevated $CO_2$ filters away all seawater bacteria before inoculating the water with non-axenic phytoplankton lab cultures. Phytoplankton culture parameters possibly selected for a fast-growing bacterial community that was adapted to phytoplankton bloom conditions and variation in water pH due to phytoplankton respiration processes. This would mean that the studied BCC was likely preconditioned to fluctuations in $CO_2$ with non-adaptive species outcompeted in semi-batch phytoplankton cultures prior to the experiment. A discussion or mentioning of this is missing. Data about other microbial measurements, such as bacterial activities or cell counts, are missing – questioning if bacterioplankton actually was the initial target of the study. Did the authors develop the network method themselves as references in the method section about networks are missing? In that case the method should have been validated. The flaws of experimental design, setup and continuous samplings are complemented by insufficiently described materials and methods.

Text and style of the manuscript are poor: several references are misplaced, missing or incorrectly cited in the reference list. The text contains word/grammar mistakes, word-autocorrect errors and the style of the text is inconsistent throughout the manuscript.

Specific comments. The title is misleading. The effects of elevated $CO_2$ on BCC were not statistically tested prior to day 6 when $CO_2$ concentration actually differed between treatments and the bacterioplankton community was artificially induced by contamination. I doubt that the authors' results support the statement "Insignificant effects of elevated $CO_2$ on bacterioplankton community in a eutrophic coastal mesocosm experiment"

Methods: page 5, line 18. What was the purpose of filtering the seawater for the mesocosms if the aim of the study was to study the bacterioplankton community? If the majority of the bacteria originated with the phytoplankton cultures, why does the community composition in Fig S.1 look very different from the community composition of the mesocosms at day4? At day4, the class distribution of LC mesocosms shows nearly 50% Epsilonbacteria in D4.1, while no Epsilonbacteria are reported from the coccolithophore or diatom cultures.

page 5, line 20. The insitu seawater pCO2 was 650 $\mu$atm. How relevant are control mesocosms where the pCO2 concentration is lowered? Despite it changing the carbon chemistry, seawater with 400 $\mu$atm seems not to reflect the eutrophic coastal environment in the Wuyuan Bay during January and is therefor a questionable control to test the hypothesis.

page 6, line3. How did the pH change over time and when were samples taken? During phytoplankton blooms, this has major importance as pH changes with respiration during the day and can shift largely over the course of 24 hours.

page 6, line 8. Mesocosms were bubbled with air containing 1000 ppm and 400 ppm CO2, yet differences in CO2 concentrations could not be maintained throughout the experiment. Why?

page 7, line 3. Can the authors show that the bacterial community composition at the beginning of the experiment was the same in all mesocosm bags? If not, their hypothesis cannot be tested!

page 7, line 14. BCC at day zero or 1 was not sampled.

page 7, line 18. Sequential filtering prior DNA extraction – missing discussion about the majority of bacteria not being included in the results (particle attached and algae associated/attached bacteria were filtered away).

page 7, line 19. Which DNA extraction protocol was used? phenol/chloroform method?

The method description is insufficient.

page 8, line 9. The QIIME pipeline is not sufficiently described. How many raw sequences were obtained? How many samples were sequenced/passed quality control? Which pipeline parameters were used? How was the phylogenetic tree produced? What kind of tree is it?

Section 2.5 is missing references, parameter description or validation of the method, the link to the sequencing center IEG is insufficient here.

Results page 10, line 11. Additional to pCO2 levels, the measured pH should be shown in a graph.

The results sections contain many passages of discussion that should not be included here (for example page 11, line 19 or page 14, line 16).

page 11, line 16. How many sequences were included in the results? How many reads were obtained per sample? Why were some replicates not included in the results?

page 12, line 20. Was the BCC tested for differences prior to day6? If so, results are not described or included in Table2.

On page 12, some bacteria phyla were selected for analysis, does it mean that the rest was ignored in analysis after this point and in the network analysis?

How similar/different are mesocosm replicates? Inter-treatment variability seems to be very high, possibly coupled to initial differences in bacterial communities in the different mesocosms.

page 14, line 12. Naming of OTUs is weird (e.g. OTU 4331023), the high numbers suggest many OTUs, but only 4992 were reported.

Can the authors support the results with bacterial abundance data? If certain bacteria increase/decrease in relative abundance, is this due to a change in community composition or an overall increase/decrease in cell numbers? This would stress the effect of

the phytoplankton bloom on bacterial growth and BCC.

Discussion The discussion is too short, selective and does not truly discuss the results in a broad perspective. For example:

Page 15, line 17. If the BCC resulted from phytoplankton culture inoculum, the bacteria were adapted to growth alongside phytoplankton in cultures and closed containers and resulting pH ranges due to phytoplankton respiration (possibly for several years, depending on when phytoplankton strains were isolated, non-adapted bacteria would have been outcompeted prior to the experiment). Therefore, the results should not be generalized but discussed in this perspective.

page 17, line 22. The authors "speculate that the stimulation of growth of Flavobacteria could have been due to the enhanced activation of proteorhodopsin under the HC treatment at the early stage of diatom bloom". This is pure speculation based only on selective reading of the literature and has no place here in the absence of any evidence of expression of proteorhodopsin.

Figures: Figure 1 is not relevant for the manuscript.

In Figure 2, SE or SD (description missing in Figure legend) should be shown both upwards and downwards.

Figure 3 misses a description of replicate numbers. Why does day 4 only have one replicate? It would aid the reader to have spaces between the different days. Inter-replicate variability is apparent, mesocosm 8 for example has a distinct BCC compared to other LC mesocosms (increase of Phaeobacter over time), however this is not discussed in the paper.

Figure 4, which information does this figure show that are not visual in Figure 3? How many replicates were included?

Figure 5, which data were used for the network? Which day/replicates? How are differences in replicate numbers accounted for? How are "OTUs with importance" eval-

[Figure]

uated?

Fig S1, how representative is the diatom BCC if it comes from two species? Is it the sum/average of cultures? Replicates? When were samples taken? During inoculation or before/after the experiment? BCC likely changes throughout the course of phytoplankton growth (as shown by the authors in the mesocosm experiment) and can affect the BCC of the inoculum.

Fig S2, the Figure text is not sufficient. How was the tree generated? What kind of tree is this? Is it rooted? Which parameters were used when it was generated? Is it relevant?

S5, the figure illustrates that the bacterioplankton diversity is widely spread in the early days of the experiment, and it is obvious that replicates at day 4 are missing. Yet a discussion of these results is missing in the text.

S6, The figure legend is misleading. The PCA legend does not show the different mesocosm replicates and they are not mentioned in the figure text. How similar are replicates (at the same day)?

---

## Referee Comment (RC2) · Anonymous Referee #2 · 25 Apr 2017

The goal of this study was to assess the effect of ocean acidification (OA) on the bacterial community during an "induced phytoplankton bloom" in a coastal area. The coastal water was filtered onto 0.1 $\mu$m (but some bacteria were present at the start of the experiment) then three xenic phytoplankton cultures were added to the mesocosms. Despite the massive sequencing work, there are important points that have not been addressed by the authors in the experimental design as well as in the sampling and analysis steps thus weakening the paper.

The authors do not show the community structure of the "contaminated water" at the beginning of the experiment (prior phytoplankton amendment) and this is a critical point in order to be able to state whether there is an effect or not of OA on bacterial community structure. It would be important to discuss how different the contaminated water community was in comparison to the bacterial community associated with the phytoplankton strains. I would encourage the authors to present also the bacterial abundance data (the authors say that bacteria were present in the "contaminated water and I assume that they have counted them) that will be very useful to understand the bacterial dynamic and response to OA. Furthermore, the DOC and POC data should be included here since the authors state that data those have been packaged in another paper.

The section Environmental parameters and experimental timeline is confusing. The authors could consider to include a table that summarizes the nutrient trends and if possible other important data (bacteria count, viral count, phytoplankton count, DOC and POC)

Some graphs in the main text and in the SI are not very informative such as phylum distribution and genus distribution graphs and confuse the message of the paper.

The SI material needs more explanation and for instance the PCA graphs do not show very clearly the findings.

It would be useful that the authors would comment the use of their primers in the light of the Environ Microbiol. 2016 May;18(5):1403-14. doi: 10.1111/1462-2920.13023. Epub 2015 Oct 14: Every base matters: assessing small subunit rRNA primers for marine microbiomes with mock communities, time series and global field samples by Parada et al.

The English and the structure of the paper should be revised.

―――――――――――――――――――――――

---

## Author Comment (AC1) · 3 Jun 2017

The manuscript addresses the research question if bacterial communities in eutrophic coastal areas will be affected by elevated CO2 concentration. The topic is highly relevant given the possible effects of changes in oceanic carbon chemistry on bacterioplankton communities and subsequent biogeochemical nutrient cycles. The authors state that they found "insignificant effects of elevated CO2 on bacterioplankton communities)", however their methodology and experimental setup is poor and insufficient to test the hypothesis. The major criticisms of the manuscript is that the bacterial community composition (BCC) resulted from contamination of tubing and material used,

as well as non-axenic phytoplankton cultures and hardly represents a natural bacterioplankton community. Even if the bacteria found in the mesocosms were of marine origin, the initial community composition is unknown and not shown to be similar among the mesocosms. Therefore the results and study are not reproducible.

Response: We appreciate the reviewers' comments and suggestions on the manuscript. Oceanic systems are open to the air with continuous exchanges of substances and microbes. In our experimental system, the mesocosms were open and aerated with filtered air of different levels of $CO_2$. Therefore, these mesocosms are subject to fluctuating environmental conditions and comparatively (relative to indoor or closed large-scale cultures) closer to natural conditions, other than the manipulated $CO_2$ levels. What we were trying to test were the basic principles of how a bacterial community changes along with phytoplankton growth under the influence of elevated $CO_2$. To investigate this, we used a model bacterial community composed of taxa originally associated with the cultured algal inoculum, combined with the natural marine assemblage that inevitably entered the mesocosms from sea spray, etc. It would have been impractical to cultivate the large volumes of axenic phytoplankton we would have needed to inoculate the mesocosms without adding any bacteria from the phytoplankton cultures. At any rate, in the end the bacterial taxa present largely resembled those found in the natural community, suggesting the resident marine bacterial assemblage was able to dominate over the added cultivated bacteria. We agree that if in situ natural phytoplankton and bacterioplankton communities were used in this mesocosm experiment, it would more closely reflect the effects of ocean acidification on the mixed natural phytoplankton and bacterioplankton communities. Considering the number of studies that have been done on the model phytoplankton responses to OA that have been carried out in laboratory, we felt it would a useful intermediate step to use model phytoplankton species to initiate the mesocosm studies before using natural communities. Therefore, we used filtered (0.01um) seawater that did not have any bacteria in all the mesocosms in the beginning. Then we inoculated phytoplankton culture containing bacterioplankton into the mesocosms. Bacterial populations developed gradually

with air-sea exchanges. We believe that using filtered seawater with inoculated isolates was reasonable and logistically practical for our experiment. Our experiment was designed as an intermediary step between laboratory and natural community field experiments, with isolates of non-axenic phytoplankton being added to filtered natural waters. In this way, we were able to investigate the effect of OA on phytoplankton and bacterioplankton in eutrophic coastal waters while minimizing the complexity of shifting compositions of natural phytoplankton communities. That is, all the mesocosms start from the same point in terms of BCC or phytoplankton composition. The correlated data on phytoplankton using this mesocosm system entitled "Carbon assimilation and losses during an ocean acidification mesocosm experiment, with special reference to algal blooms" will soon be published at Marine Environmental Research (in press). BCC in our study could be the combined result of a combination of the inoculated phytoplankton, air-sea exchange and sampling. Previous mesocosm experiments started with natural communities also had BCC from air-change and sampling. The important point is that each mesocosm has the same BCC, as in previous mesocosm studies. The dynamics of bacterioplankton throughout previous mesocosm studies were also due to the combination of the original bacterioplankton community added in the mesocosm bags in the beginning and any outside bacterioplankton that entered during the experiment.. Furthermore, bacteria were not detectable by flow cytometry in the filtered seawater just before inoculation. Three species of non-axenic phytoplankton with bacterioplankton were mixed and then inoculated into each mesocosm bag. So the initial bacterioplankton community was considered the same among all mesocosms. We revised the manuscript and double checked the data and their interpretations to further explained the reasons that we used filtered seawater for our eutrophic coastal seawater mesocosm experiment as well as the strengths and weaknesses of this experimental design.

In fact, samples of the initial days are missing. The BCC after 4 days looks different between mesocosms, yet 3 replicates are missing in the figures, results section and statistical analysis without mentioning.

Response: We tried to do sampling at day 2 but the samples were not successfully collected, probably due to very high concentration of TEP (Transparent Exopolymer Particles) which easily blocked the polycarbonate filter for bacterioplankton collection. According to the bacterioplankton abundance data in Yibin Huang et al (entitled "responses of phytoplankton and bacterial metabolism to CO2 enrichment during a coastal mesocosm experiment", under revision after first-round review for Limnology and Oceanography), the bacterioplankton abundance was very high at day 2 and day 4 which may be associated with high TEP concentration (Sugimoto et al., 2007, Ramaiah et al., 2000). We also tried to do sampling at day 4. But eventually we successfully extract enough DNA for sequencing only from bag 1, bag 7 and bag 6. So some replicates were missing in the Figure 3. The replicates of HC and LC were mentioned in material and method section (Page 6 line 6-7). The replicates have been mentioned again in statistical analysis, result section and figure legends to make it easier for the readers.

Generally, it appears bizarre that a study addressing the BCC response to elevated CO2 filters away all seawater bacteria before inoculating the water with non-axenic phytoplankton lab cultures. Phytoplankton culture parameters possibly selected for a fast-growing bacterial community that was adapted to phytoplankton bloom conditions and variation in water pH due to phytoplankton respiration processes. This would mean that the studied BCC was likely preconditioned to fluctuations in CO2 with non-adaptive species outcompeted in semi-batch phytoplankton cultures prior to the experiment. A discussion or mentioning of this is missing.

Response: This is a very good point. We agree that the bacterioplankton originated from phytoplankton culture likely outcompeted other non-adaptive species in semi-batch phytoplankton cultures prior to the experiment. We have added some sentences in the discussion to address this point (Page 17 Line 6-9).

Data about other microbial measurements, such as bacterial activities or cell counts, are missing – questioning if bacterioplankton actually was the initial target of the study.

Did the authors develop the network method themselves as references in the method section about networks are missing? In that case the method should have been validated.

Response: Bacterial activities and bacterial cell abundance data were shown in another paper (Yibin Huang et al, under revision of Limnology and Oceanography). We did not develop the network analysis method by ourselves. We followed the network construction methodology described in Wang et al., 2016. The reference for network construction and analysis has been added to the method and material section (Page 10 Line 12).

The flaws of experimental design, setup and continuous samplings are complemented by insufficiently described materials and methods. Text and style of the manuscript are poor: several references are misplaced, missing or incorrectly cited in the reference list. The text contains word/grammar mistakes, wordautocorrect errors and the style of the text is inconsistent throughout the manuscript.

Response: We improved the materials and method section to clarify the experimental design and sampling. The references have been rearranged carefully. The text has been revised carefully and the English has been polished.

Specific comments. The title is misleading. The effects of elevated $CO_2$ on BCC were not statistically tested prior to day 6 when $CO_2$ concentration actually differed between treatments and the bacterioplankton community was artificially induced by contamination. I doubt that the authors' results support the statement "Insignificant effects of elevated $CO_2$ on bacterioplankton community in a eutrophic coastal mesocosm experiment".

Response: We agree that if the data prior to day 6 were shown in the manuscript, the conclusion would be more solid. It's a pity that we only successfully obtain several samples for sequencing at day 4 due to the reasons mentioned above. The pH values were statistically different from day 0 to day 10. So our results and analysis were still

meaningful. Although the pH was maintained at the target pH value throughout the experiment, this doesn't mean that all the results based on mesocosm experiments were meaningless. In the natural environment, pH increases gradually throughout the phytoplankton bloom. Our experiment and previous mesocosm experiments could be considered as the phytoplankton bloom initiated with different CO2 concentration/pH.

Methods: page 5, line 18. What was the purpose of filtering the seawater for the mesocosms if the aim of the study was to study the bacterioplankton community?

Response: As mentioned above, we wanted to minimize the complexity of shifting compositions of natural phytoplankton communities and using filtered seawater was reasonable and practical for our eutrophic coastal seawater mesocosm experiment. Furthermore, according our unpublished data, the bacterioplankton in phytoplankton cultures played important roles under ocean acidification which were usually ignored in previous studies. So we think the effects of ocean acidification on bacteriplankton in phytoplankton cultures is worth to be investigated in a larger scale experiment, which was our original purpose. However, as noted above the bacterioplankton from natural environment gradually became dominant in the mesocosm bags. So actually, the bacterioplankton we studied in this paper were mainly bacterioplankton from the natural environment.

If the majority of the bacteria originated with the phytoplankton cultures, why does the community composition in Fig S.1 look very different from the community composition of the mesocosms at day4? At day4, the class distribution of LC mesocosms shows nearly 50% Epsilonbacteria in D4.1, while no Epsilonbacteria are reported from the coccolithophore or diatom cultures.

Repsonse: The results suggest that the outside bacterioplankton replaced the bacteria originating in the phytoplankton culture and became the dominant bacterioplankton in the mesocosm over day 0 to day 4. So Fig S.1 looks very different from the community composition of the mesocosms at day 4.

page 5, line 20. The in situ seawater pCO2 was 650 uatm. How relevant are control mesocosms where the pCO2 concentration is lowered? Despite it changing the carbon chemistry, seawater with 400 ïA■atm seems not to reflect the eutrophic coastal environment in the Wuyuan Bay during January and is therefore a questionable control to test the hypothesis.

Response: We agree that 400 uatm may not reflect the eutrophic coastal environment in the Wuyuan Bay during January. However, the system we used was an intermediary step between laboratory and natural community, not a natural environment experimental system even though filtered eutrophic seawater was used. So the bigger contrast between control (400 uatm) and treatment (1000 uatm) was used for us to better observe the effects of elevated CO2. So we suggest that choosing 400 uatm as the control in our study was reasonable.

page 6, line3. How did the pH change over time and when were samples taken? During phytoplankton blooms, this has major importance as pH changes with respiration during the day and can shift largely over the course of 24 hours.

Response: The samples in this study were collected at about 10 am each time while the other parameters were also measured simultaneously. We agree that the pH variation over the course of 24 hours should be considered during the phytoplankton blooms. It was pity that we did not collect bacterioplankton samples over the course of 24 hours. The comment "In future studies, it would be also worthwhile to sample over a diel cycle in order to understand the cyclic variability in pH and whether this affects short term changes in bacterioplankton community structure." has been added in the discussion section (Page 21 Line 21-22).

page 6, line 8. Mesocosms were bubbled with air containing 1000 ppm and 400 ppm CO2, yet differences in CO2 concentrations could not be maintained throughout the experiment. Why?

Response: When phytoplankton bloom occurred and phytoplankton cells reached high

concentration, the consumption of CO2 was much higher than during the early stage. So this meant that the CO2 concentrations could not be maintained when phytoplankton entered into logarithmic growth stage. For indoor semi continuous ocean acidification experiments with CO2 bubbling, the cultures have to be diluted periodically to maintain the cell concentration and thus control the CO2 concentration. But such dilution was not possible in this mesocosm experiment considering the big volume of seawater in each mesocosm bag.

Page 7, line 3. Can the authors show that the bacterial community composition at the beginning of the experiment was the same in all mesocosm bags? If not, their hypothesis cannot be tested! page 7, line 14. BCC at day zero or 1 was not sampled.

Response: At the beginning of this experiment, no bacteria were detected prior to phytoplankton inoculation. The phytoplankton culture with bacterioplankton were evenly distributed into each bags for inoculation. So we considered the bacterial community composition at the beginning of the experiment was the same or similar in all mesocosm bags. As for day 0, no detectable bacterioplankton were detected before inoculation. We agree that it is better to show the data at day 2, but unfortunately we were unable to collect samples due to the technical limitations mentioned above.

page 7, line 18. Sequential filtering prior DNA extraction – missing discussion about the majority of bacteria not being included in the results (particle attached and algae associated/attached bacteria were filtered away).

Response: We agree with you that the majority of the particle attached and algae attached bacteria were filtered out by sequential filtering. Consequently, the bacterioplankton in our study did not include these bacteria. This has been added to the discussion section (Page 21 Line 12-13).

page 7, line 19. Which DNA extraction protocol was used? phenol/chloroform method?

Response: The detailed DNA extraction protocol: 1. Wash the filter with 1 ml of lysis

buffer described in (Francis et al., 2005) and 10 ul of lysozyme (100 mg/ml), vortex and incubate at 37 degrees for 30 minutes. 2. Add 5 ul RNase A (10 mg/ml), incubate at 37 degrees for 30 minutes. 3. Add 20 ul proteinase K 4. Add 220 ul GB solution from Bacteria DNA extraction kit (Tiangen DP302) 5. Follow the Bacteria DNA extraction kit's instruction to finish the DNA extraction.

The method description is insufficient. page 8, line 9. The QIIME pipeline is not sufficiently described. How many raw sequences were obtained? How many samples were sequenced/passed quality control? Which pipeline parameters were used? How was the phylogenetic tree produced? What kind of tree is it? Section 2.5 is missing references, parameter description or validation of the method, the link to the sequencing center IEG is insufficient here.

Response: When the sequencing finished, we need to filter the raw data to secure the quality of our data, which mainly including: 1) Cut the polluted adapter; 2) Remove low quality reads, specifically reads with average quality less than 19, based on the Phred algorithm; 3) Remove the reads with N base exceeding 5%. Finally 2972070 raw reads were obtained in total from all the samples and 2365844 reads passed quality control (see Supplementary Table 1), the average of clean read rate was 79.65%. According to the reference database, the representative sequences for each OTU were aligned using PyNAST (Caporaso et al., 2010), finally the phylogenetic tree was generated from the Graphlan (Langille et al., 2013) using information on both the relative abundance and phylogenetic relationship of observed species. The missing references have been added to the method section (Page 14 Line 14-22).

Results page 10, line 11. Additional to pCO2 levels, the measured pH should be shown in a graph. The results sections contain many passages of discussion that should not be included here (for example page 11, line 19 or page 14, line 16). page 11, line 16.

Response: The pH value has been added in Figure 2 with pCO2 levels. The results section that contained passages of discussion has been moved to the discussion section or rephrased. The structure of this manuscript has been rearranged.

How many sequences were included in the results? How many reads were obtained
per sample? Why were some replicates not included in the results? page 12, line 20.
Was the BCC tested for differences prior to day6? If so, results are not described or
included in Table 2.

Response: The raw reads and the clean reads of each sample were shown in supple-
mentary table 2. As mentioned above, probably due to high concentration of TEP, all
the samples at day 2 were not successfully collected and only a few samples at day 4
were successfully collected probably.

On page 12, some bacteria phyla were selected for analysis, does it mean that the
rest was ignored in analysis after this point and in the network analysis? How sim-
ilar/different are mesocosm replicates? Inter-treatment variability seems to be very
high, possibly coupled to initial differences in bacterial communities in the different
mesocosms.

Response: All the bacteria phyla were analyzed in the network analysis. We agreed
that inter-treatment variability was high. This mesocosm experiment was conducted
outdoors and the mesocosm enclosures were exposed to fluctuating environmental
factors which led to high inter-treatment variability. Previous mesocosm experiments
also have similarly high inter-treatment variability, which is very hard to avoid for out-
door mesocosm experiments. We did sampling every two days which also can intro-
duce outside bacteria randomly. So we think the high inter-treatment variability was
due to the mesocosm experiment itself, rather than to initial differences in bacterial
communities in the different mesocosms.

page 14, line 12. Naming of OTUs is weird (e.g. OTU 4331023), the high numbers sug-
gest many OTUs, but only 4992 were reported. Can the authors support the results with
bacterial abundance data? If certain bacteria increase/decrease in relative abundance,
is this due to a change in community composition or an overall increase/decrease in
cell numbers? This would stress the effect of the phytoplankton bloom on bacterial growth and BCC.

Response: The OTU IDs in our study were IDs in Greengene database. The increase/decrease of certain bacteria in relative abundance is due to a change in community composition, not an overall increase/decrease in cell numbers. There was no big variation in the cell density from Day 12 to Day 32 according to Yibin HUANG et al (Limnology and Oceanography, under revision). However, our data showed a big variation in community composition between day 13 and day 29. All above information indicated that bacteria increase/decrease in relative abundance was due to the change in community composition, not the overall increase/decrease in cell numbers.

The discussion is too short, selective and does not truly discuss the results in a broad perspective. For example: Page 15, line 17. If the BCC resulted from phytoplankton culture inoculum, the bacteria were adapted to growth alongside phytoplankton in cultures and closed containers and resulting pH ranges due to phytoplankton respiration (possibly for several years, depending on when phytoplankton strains were isolated, non-adapted bacteria would have been outcompeted prior to the experiment). Therefore, the results should not be generalized but discussed in this perspective.

Response: We agree with the reviewer that the inoculated bacterioplankton along with the phytoplankton probably have outcompeted the non-adapted bacteria prior to the experiment. It seems though that the environmental bacterioplankton from outside through tubes, sampling and sea air exchange became dominant in the mesocosms from day 0 to day 4, because the bacterioplankton composition at day 4 and day 6 were very different from the bacterioplankton composition in the original phytoplankton cultures, including some which were not detected in the phytoplankton cultures at all. This suggests the local bacterioplankton outcompeted the bacterioplankton from the phytoplankton cultures at an early stage of the mesocosm experiment. Everything mentioned above has been added to the discussion section. Because of this shift to natural bacteria, we think the results about the bacterioplankton community composition under the HC and LC conditions can be generalized, as on Page 15, line 17.

page 17, line 22. The authors "speculate that the stimulation of growth of Flavobacteria could have been due to the enhanced activation of proteorhodopsin under the HC treatment at the early stage of diatom bloom". This is pure speculation based only on selective reading of the literature and has no place here in the absence of any evidence of expression of proteorhodopsin.

Response: We agree that this is just speculation without proteorhodopsin expression data in our study. We have rephrased this description.

Figures: Figure 1 is not relevant for the manuscript.

Response: We think showing the location of the experiment site is important for the whole manuscript. We want to show Wuyuan Bay is in the city center and strongly influenced by human activity. To address this comment though, this figure has been moved to supplementary data.

In Figure 2, SE or SD (description missing in Figure legend) should be shown both upwards and downwards.

Response: SD with upwards and downwards has been added in Figure 2. The description of SD has been added in the Figure 2 legend as well.

Figure 3 misses a description of replicate numbers. Why does day 4 only have one replicate? It would aid the reader to have spaces between the different days. Interreplicate variability is apparent, mesocosm 8 for example has a distinct BCC compared to other LC mesocosms (increase of Phaeobacter over time), however this is not discussed in the paper.

Response: The replicate numbers have been added in the Figure 3 legend. As mentioned above, we tried to collect the samples and extract DNA from all mesocosm bags but we only successfully extracted enough DNA from bag 1 and bag 6 at day 4 for sequencing. Extra space between different days have been added in Figure 3. We

agree that mesocosm 8 has distinct BBC compared to the other LC mesocosms. We think the high inter-replicate variability was due to the experimental environment. The increase of Phaeobacter in mesocosm 8 was a random issue in this mesocosm experiment. The discussion about the distinct BBC in mesocosm 8 has been added in the discussion (Page 17 Line 19-22).

Figure 4, which information does this figure show that are not visual in Figure 3? How many replicates were included?

Response: Figure 3 showed the overview of community structure at different taxonomic levels of all the samples. But it is not easy to get information about the change of certain bacteria groups throughout the experiment. Figure 4 showed clearly the change of Bacteroidetes in contrast with Proteobacteria at the phylum level; Flavobacteria in contrast with Alphabacteria at the class level; Flavobacteriales in contrast with Rhobacteriales at the order level; and Flavobacteriaceae in contrast with Rhodobacteriaceae at the family level. 3 replicates were included except the missing samples at day 4 and day 6 for Figure 3 and Figure 4.

Figure 5, which data were used for the network? Which day/replicates? How are differences in replicate numbers accounted for? How are "OTUs with importance" evaluated?

Response: We used all the data we have from each bag on each day, except some samples that were missing on day 4 and day 6 for network analysis. The sequencing data from each mesocosm bag throughout the experiment at different time points were considered as different replicates with time series. For example, the sequencing data from mesocosm bag 1 with time series at day 4, day 6, day 8, day 19 and day 29 were considered as HC1. Mesocosm 1, 6 and 8 were three replicates for HC treatment and mesocosm 2, 4 and 7 were three replicates for LC treatment. The main text about network construction in method and material section has been revised as "Firstly, the similarity matrices of the relative abundance of OTUs in LC and HC conditions were

created respectively using Pearson correlation coefficient across time points with biological replicates by a random matrix theory (RMT)-based approach". OTUs with high relative abundance were defined as OTUs with importance. OTU 572670 with 21402 reads from all the samples, OTU 558885 with 5780 reads, OTU 190052 with 42525 reads, OTU107130 with 12892 reads, OUT 572670 with 21402 reads, OUT 4331023 with 7845 reads were considered as OTUs with importance (see supplementary table 2)

Fig S1, how representative is the diatom BCC if it comes from two species? Is it the sum/average of cultures? Replicates? When were samples taken? During inoculation or before/after the experiment? BCC likely changes throughout the course of phytoplankton growth (as shown by the authors in the mesocosm experiment) and can affect the BCC of the inoculum.

Response: The diatom BCC came from the sum of two species of culture. The phytoplankton culture samples were taken after the inoculation in order to investigate the roles of phytoplankton culture BCC in the whole mesocosm experiment. It cannot be denied that it would have been better to collect the bacterioplankton from the phytoplankton just before inoculation. We think the BCC of phytoplankton culture should be stable over the short term, because the phytoplankton cultures were maintained in semi-continuous culture with artificial seawater.

Fig S2, the Figure text is not sufficient. How was the tree generated? What kind of tree is this? Is it rooted? Which parameters were used when it was generated? Is it relevant?

Response: PyNAST method (Caporaso, et al.,2010) and Graphlan software (Langille, et al., 2013) were used to construct the phylogenetic unrooted NJ tree as mentioned above. The legend of Fig S2 has been revised.

S5, the figure illustrates that the bacterioplankton diversity is widely spread in the early days of the experiment, and it is obvious that replicates at day 4 are missing. Yet a

discussion of these results is missing in the text.

Response: The explanation of missing data at day 4 has been mentioned above, and added in the methods and materials section.

S6, The figure legend is misleading. The PCA legend does not show the different mesocosm replicates and they are replicates (at the same day)?

Response: The legend of Fig. S6 has been revised to clarify that each symbol presents the average value of the HC and LC treatments with three replicates at different days. For example, HC-D13 presents the average value of HC2, HC4, HC7 at day13.

Reference:

Caporaso, J. G., Bittinger, K., Bushman, F. D., Desantis, T. Z., Andersen, G. L., and Knight, R. 2010. PyNAST: A flexible tool for aligning sequences to a template alignment. Bioinformatics, 26: 266–267.

Francis, C. A., Roberts, K. J., Beman, J. M., Santoro, A. E., and Oakley, B. B. 2005. Ubiquity and diversity of ammonia-oxidizing archaea in water columns and sediments of the ocean, 102: 14683–14688.

Langille, M., Zaneveld, J., Caporaso, J. G., McDonald, D., Knights, D., Reyes, J., Clemente, J., et al. 2013. Predictive functional profiling of microbial communities using 16S rRNA marker gene sequences. Nature biotechnology, 31:81421.

Ramaiah, N., Sarma, V. V. S. S., Gauns, M., Dileep Kumar, M., and Madhupratap, M. 2000. Abundance and relationship of bacteria with transparent exopolymer particles during the 1996 summer monsoon in the Arabian Sea. Proceedings of the Indian Academy of Sciences, Earth and Planetary Sciences, 109: 443–451.

Sugimoto, K., Fukuda, H., Baki, M. A., and Koike, I. 2007. Bacterial contributions to formation of transparent exopolymer particles (TEP) and seasonal trends in coastal waters of Sagami Bay, Japan. Aquatic Microbial Ecology, 46: 31–41.

Wang, Y., Zhang, R., Zheng, Q., Deng, Y., Van Nostrand, J. D., Zhou, J., and Jiao, N. 2016. Bacterioplantkon community resilience to ocean acidification: evidence from microbial network analysis. ICES J. Mar. Sci., 73: 865–875.
* * *

---

## Author Comment (AC2) · 3 Jun 2017

The goal of this study was to assess the effect of ocean acidification (OA) on the bacterial community during an "induced phytoplankton bloom" in a coastal area. The coastal water was filtered onto 0.1 um (but some bacteria were present at the start of the experiment) then three xenic phytoplankton cultures were added to the mesocosms. Despite the massive sequencing work, there are important points that have not been addressed by the authors in the experimental design as well as in the sampling and analysis steps thus weakening the paper. The authors do not show the community structure of the "contaminated water" at the beginning of the experiment (prior phytoplankton amendment) and this is a critical point in order to be able to state whether there is an effect or not of OA on bacterial community structure. It would be important to discuss how different the contaminated water community was in comparison to the bacterial community associated with the phytoplankton strains.

Response: We appreciate the comments from reviewer #2. The description of the experimental design, sampling and analysis have been strengthened in the revised manuscript. Our experiment was designed as an intermediary step between laboratory and natural community field experiments, with isolates of non-axenic phytoplankton being added to filtered natural waters. In this way, we were able to investigate the effect of OA on phytoplankton and bacterioplankton in eutrophic waters while minimizing the complexity of shifting compositions of natural phytoplankton communities. In other words, we aimed to study the effects of ocean acidification on some model phytoplankton species and phytoplankton culture-originated bacterioplankton in a larger scale experiment compared to the lab experiment. Therefore, this experiment could not truly reflect the effects of ocean acidification on field natural phytoplankton and bacterioplankton communities. The outdoor mesocosm system was not sterile, and it was impossible to avoid the bacteria from outside through sampling and air-sea exchange during the experiment. Our data showed that the local bacterioplankton communities were very different from bacterioplankton originated from phytoplankton culture by day 4 based, on the comparison of the bacterioplankton community at day 4 and the original bacterioplankton community. And some bacterioplankton that were not detected in the original phytoplankton culture appeared in samples collected at day 4. Therefore, we conclude that the environmental bacterioplankton outcompete the phytoplankton-originated bacterioplankton from day 0 to day 4. Since the day 2 data were lacking, it seems likely that the environmental bacterioplankton became dominant even before day 4. This suggests the bacterioplankton studied in this paper were mainly natural bacterioplankton. The points mentioned above have been added to the results and discussion section. We agree that it is important to discuss the contaminated water community in comparison to the bacterial community associated with the phytoplankton by showing the bacterial community structure at day 2 and day 4. We tried to do sampling at day 2 and day 4. But eventually we could successfully extract enough DNA only from bag 1, bag 6 and bag 7 at day 4 for sequencing, probably due to high concentration of TEP (Transparent Exopolymer Particles) (Sugimoto et al., 2007, Ramaiah et al., 2000). Bacteria were not detectable by flow cytometry in the filtered seawater prior to inoculation. Three species of non-axenic phytoplankton with bacterioplankton were mixed and then inoculated into each mesocosm bag. Because the mixture added was the same, we considered the initial bacterioplankton community was similar in each mesocosm bag. We described the experimental design in a more detailed way to clarify why we used this approach in the revised manuscript. The limitations of our experimental design and approaches were also pointed out in the manuscript (Page 7 Line 16-22, Page 8 Line 1-5).

I would encourage the authors to present also the bacterial abundance data (the authors say that bacteria were present in the "contaminated water and I assume that they have counted them) that will be very useful to understand the bacterial dynamic and response to OA. Furthermore, the DOC and POC data should be included here since the authors state that data those have been packaged in another paper.

Response: We agree that it is better to discuss the correlation between bacterioplankton abundance and community structure in the manuscript as well as DOC and POC data in this paper. The bacteria abundances were shown in Yibin Huang et al entitled "responses of phytoplankton and bacterial metabolism to CO2 enrichment during a coastal mesocosm experiment" (in the second round revision at Limnology and Oceanography). DOC and POC data were shown in Nana Liu et al (in press at Marine Environmental Research) . The section Environmental parameters and experimental timeline is confusing. The authors could consider to include a table that summarizes the nutrient trends and if possible other important data (bacteria count, viral count, phytoplankton count, DOC and POC) RE: Sorry for the confusion. We agree that the nutrient trends, bacteria abundance data, phytoplankton abundance data, DOC and

POC data are important for supporting our main results (Viral counting was not done in this mesocosm experiment). However, these data were packaged in other papers either published or under revision as mentioned above. We think it is not appropriate to use these data directly in this paper. We have cited these papers containing bacteria counts, phytoplankton counts, and DOC and POC data.

Some graphs in the main text and in the SI are not very informative such as phylum distribution and genus distribution graphs and confuse the message of the paper. The SI material needs more explanation and for instance the PCA graphs do not show very clearly the findings.

Response: We improved the legends of the supplementary figures and the text to make them more informative. For example, the software used to construct the phylogenetic tree and the type of phylogenetic tree has been added into the legend of Fig.S2. The explanation of different replicates of the HC and LC treatment has been clarified in the legend Fig. S6 (PCA graph).

It would be useful that the authors would comment the use of their primers in the light of the Environ Microbiol. 2016 May;18(5):1403-14. doi:10.1111/1462-2920.13023. Epub 2015 Oct 14: Every base matters: assessing small subunit rRNA primers for marine microbiomes with mock communities, time series and global field samples by Parada et al.

Response: The choice of primers amplifying 16S genes is crucial. Sequencing depth, high coverage of the taxa of interest, the ability to compare results with prior studies, accuracy in relative abundances and the phylogenetic resolution of the sequenced PCR products should be considered when choosing suitable primers (Parada et al., 2016) We used primers 341F (5'-CCTACGGGNGGCWGCAG-3') and 805R (5'-GACTACHVGGGTATCTAATCC-3') targeting the 16S V3-V4 region, which has successfully been applied in previous studies (Hugerth et al., 2014). Thus we used 341F/805R primers that were well accepted for bacteria diversity studies. For our

study, using 341F/805R was appropriate considering the ability to compare results with prior studies, accuracy in relative abundances and the phylogenetic resolution of the sequenced PCR products. The paper mentioned above mainly discussed about the primers 515F-Y/926R and 515F-C/806R targeting the 16S V4-V5 region. The advantage of these two pairs of primers is that it should match bacteria as well as archaea. Therefore, the archaea were missing in our data set based on the primers 341F/805R we used in this study. We think primers 515F-Y/926R are better candidates because of their better coverage and their sequences have been validated in Parada et al. Thus we think 515F-Y/926R will be useful for future bacteria diversity studies. The limitations of the primers used in this study has been added to the discussion section (Page 21 Line 13-14).

The English and the structure of the paper should be revised.

Response: The text and the structure has been revised carefully.

References: Hugerth, L. W., Wefer, H. A., Lundin, S., Jakobsson, H. E., Lindberg, M., Rodin, S., Engstrand, L., et al. 2014. DegePrime, a program for degenerate primer design for broad-taxonomic-range PCR in microbial ecology studies. Applied and Environmental Microbiology, 80: 5116–5123.

Parada, A. E., Needham, D. M., and Fuhrman, J. A. 2016. Every base matters: Assessing small subunit rRNA primers for marine microbiomes with mock communities, time series and global field samples. Environmental Microbiology, 18: 1403–1414.

Ramaiah, N., Sarma, V. V. S. S., Gauns, M., Dileep Kumar, M., and Madhupratap, M. 2000. Abundance and relationship of bacteria with transparent exopolymer particles during the 1996 summer monsoon in the Arabian Sea. Proceedings of the Indian Academy of Sciences, Earth and Planetary Sciences, 109: 443–451.

Sugimoto, K., Fukuda, H., Baki, M. A., and Koike, I. 2007. Bacterial contributions to formation of transparent exopolymer particles (TEP) and seasonal trends in coastal

waters of Sagami Bay, Japan. Aquatic Microbial Ecology, 46: 31–41.

---

## Author Response (AR3)

What is still missing is an estimate on the ratio of bacteria being continuously introduced to actual standing stocks in the mesocosms. This is critical as only a low ratio would allow to detect potential $CO_2$ effects. To do this you could us maximum growth rates reported in the literature for the most abundant groups and then estimate how many you would have needed to introduce after one day of aeration on day 1 to reach the numbers you have measured on day 2. And such type of analysis would need to be thoroughly discussed in your manuscript.

RE: Assuming the bacterioplankton concentration at day 2 representing the concentration of Pseudomonadaceae, one of the most abundant bacterioplankton groups from surrounding seawater, the concentration of Pseudomonadaceae at day 1 could be estimated based on the growth rate ($\mu_{max}$ $h^{-1}$=0.75) of *Pseudomonas aeruginosa* reported in (Adav and Lee, 2008) and the bacterioplankton concentration at day 2. The estimated concentration of Pseudomonadaceae at day 1 was 101.93 cells/ml. Therefore, the ratio of bacteria being continuously introduced to actual standing stocks in the mesocosms was low, which allowed us to detect potential $CO_2$ effects in this mesocosm experiment (Page 18 , Line 4-10).

**The derivation procedure is as follows:**

$\mu$ max ($h^{-1}$)=0.75

$\mu$ max ($h^{-1}$) = (Ln X- Ln Y)/48 hours

Y: the estimated concentration of Pseudomonadaceae at day 1 based on growth rate of *Pseudomonas aeruginosa* ($\mu_{max}$ ($h^{-1}$)=0.75).

X= $6.693 \times 10^9$ cells/ml (the average bacterioplankton concentration at day 2)

A=X/$e^{48\mu}$=101.93 cells/ml

**Reference:**

[revised manuscript text omitted]

Alphaproteobacteria
Chloroplast
Gamaproteobacteria
Flavobacteriia
Betaproteobacteria
Saprospirae
Epsilonproteobacteria
Phycisphaerae

**Figure 4**

[Figure]

**Figure 5**

**Table 1** Topological properties of the bacterioplankton communities as represented by molecular networks under HC and LC treatments; also their rewired random networks.

| | Experimental network | | | | | | | Random network | | |
|---|---|---|---|---|---|---|---|---|---|---|
| | Total nodes | Total links | R2 of power-law | Average clustering coefficient (avgCC) | Average connectivity | Harmonic geodesic distance (HD) | Modularity | Average clustering coefficient (avgCC) | Harmonic geodesic distance (HD) | Modularity |
| LC | 85 | 209 | 0.817 | 0.402 | 0.625 | 3.397 | 0.414 | 0.424 +/- 0.023 | 2.187 +/- 0.049 | 0.249 +/- 0.010 |
| HC | 96 | 310 | 0.817 | 0.448 | 0.714 | 2.956 | 0.303 | 0.292 +/- 0.023 | 2.306 +/- 0.059 | 0.323 +/- 0.008 |

**Table 2** Dissimilarity tests of bacterial communities in the HC and LC treatments at various time points.

| | Anosim | | MRPP | | Adonis | |
|---|---|---|---|---|---|---|
| Time | R | P-value | $\delta$ | P-value | $R^2$ | P |
| day6 | -0.111 | 0.602 | 0.3952 | 1 | 0.15447 | 1 |
| day8 | 0.111 | 0.284 | 0.438 | 0.6 | 0.2 | 0.5 |
| day10 | 0.037 | 0.613 | 0.4929 | 0.7 | 0.17829 | 0.7 |
| day13 | 0.111 | 0.309 | 0.412 | 0.5 | 0.19714 | 0.5 |
| day19 | 0 | 0.693 | 0.4336 | 0.3 | 0.28263 | 0.3 |
| day29 | -0.259 | 1 | 0.4513 | 0.9 | 0.15517 | 0.9 |

---

## Author Response (AR4)

Dear Dr. Schulz:

We would like to thank you for your editorial handling of our paper (Interactive network configuration maintains bacterioplankton community structure under elevated $CO_2$ in a eutrophic coastal mesocosm experiment, bg-2017-10). In response to your following comments, we have looked into literatures and performed further analyses. The responses are placed under the following comments.

Associate Editor Decision: Reconsider after major revisions (24 Oct 2017) by Kai G. Schulz

Comments to the Author:

Dear authors, thank you for your reply. I am, however, not convinced by your calculations. Given the unproportional influence of a growth rate estimate on the number of bacteria that would have been introduced on day one, a more conservative approach should have been taken.

For instance, simply assuming growth rates to be half of those maximum ones you have chosen to use, I calculate a daily bacteria influx from outside the mesocosms of about 825.000 cells/ml (in comparison to about 100!). Although this should still not pose a problem to your experimental setup, the growth rate of 0.75 h-1 taken from Adave et al. (2008) for Pseudomonas aeruginosa appears to be the maximum growth rate at a pH of 7 at 35 degrees Celsius, which is far from the conditions during your incubations.

Furthermore, it appears that growth rates for marine bacteria are typically rather on the order of per day than per hour (compare for instance Kirchman 2016). This in turn would mean that the contamination of the system with bacteria from the outside would have been substantial, potentially compromising your data and its interpretation.

As it stands, I have unfortunately no choice but to reject your manuscript, unless you can provide convincing evidence or reference for such high growth rates in your bacterial community (e.g.bacterial production rate measurements).

Sincerely,

Kai Schulz

D. Kirchman, Growth rates of microbes in the Oceans, Annual Review of Marine Science, 8:4.1-4.25 (2016).

Responses:

We agree with your comment that choosing the growth rate of 0.75 h$^{-1}$ only based on (Adav and Lee, 2008) for *Pseudomonas aeruginosa* with the maximum growth rate at a pH of 7 at 35°C, is not appropriate. So we searched literatures to figure out the reported ranges in different waters or regions.

1. The bacterial growth rate could be as high as 30 d$^{-1}$ in eutrophic Bietri Bay (Paul A. White, Jacob Kalff, 1991), and bacterial growth rate reached 16.2 day$^{-1}$ (0.675 h$^{-1}$) during a diatom bloom in a mesocosm experiment using seawater from Santa Barbara Channel amended with nutrients (Smith et al., 1995). This value is near the value we used. It has been recognized that bacterial growth rates in the eutrophic coastal waters much higher than in pelagic oligotrophic waters (Kirchman, 2016). Under simulated eutrophication conditions (Philippe and Al, 1999, Table 1), the growth rate of bacteria from the Mediterranean sea ranged from 0.245 h$^{-1}$ to 0.853 h$^{-1}$ based on the data measured roughly every 24 hrs in batch mesocosms (Philippe and Al, 1999, Table 4). The bacteria growth data in the review by Kirchman (2016) were mainly based on investigations in oligotrophic seawaters with limited literatures from the eutrophic coastal waters (Kirchman, 2016, appendix 1-4).

In summary, bacterial growth rates under eutrophic conditions are much higher than under oligotrophic conditions, and it has been reported to be on the order of per hour. Our mesocosm experiment was conducted in eutrophic coastal waters with initial DOC and nitrate levels of about 260 μmol/L and 73 μmol/L respectively. Therefore, it is not unusual to expect high growth rates of bacterioplankton in our mesocosms. In addition, since top-down biotic controls can regulate bacterial abundance, presence of grazers or virus could affect the measured rates of growth (Worden et al., 2015). In our mesocosm experiments, the seawater was filtered (0.01 μm, with German-made cartridge filter) before being simultaneously pumped to all bags. Zooplankton, ciliates and flagellates were not detected throughout the experiment except Mesocosm 8, which was contaminated by dinoflagellates at the end of the experiment. Although the viral biomass was not measured, viral biomass at the beginning of the experiment should be low. Subsequently, the diatom-dominated mesocosms with eutrophic seawater, with low bacterial loss pressure, can be expected to see high bacterial growth rates.

2. Bacterial growth rate varies greatly according to different groups. *Vibrio natriegens* is a fast growing marine bacteria with growth rate of 140 d$^{-1}$, whereas *Pelagibacter ubique*, a representative of the most abundant bacterial clade in the ocean, SAR11, has growth rates of 0.4–0.6 d$^{-1}$ in laboratory pure cultures (Kirchman, 2016). Some bacteria species belong to *Pseudomonas* group were reported to have high growth rates. The growth rate of *Pseudomonas aeruginosa* was reported as high as 0.75 h$^{-1}$ (Adav, 2008) while *Pseudomonas natriegens* was also reported to have a generation less than 10 minutes (growth rate 4.33 h$^{-1}$)(Eagon, 1962). Although these data were obtained under optimal conditions in the lab, these results showed that these two species have high growth rate property.

Unfortunately, the bacterial production rate, which could reflect the growth rates of bacterioplankton community, was not measured from day 0 to day 4. So it is a pity that we have no bacterial production data to support the high growth rate in the bacterial community from day 0 to day 4. And we would like to point out that the bacterioplankton introduction from outside started from the beginning of the experiment. So we think that we should estimate the bacterioplankton abundance on day 0, instead on day 1 based on the growth rate and bacterioplankton abundance on day 2. Although it is difficult for us to accurately estimate the growth rates of the bacterioplankton (such as Pseudomonas group) introduced from outside, we assume choosing 0.5 $h^{-1}$ as the growth rate of the bacterioplankton from outside is reasonable based on the documented data mentioned above.

The bacterial growth rates under eutrophic conditions are much higher than under oligotrophic conditions (White et al., 1991; Kirchman, 2016). The bacterial growth rate reached 16.2 $day^{-1}$ (0.675 $h^{-1}$) during a diatom bloom in a mesocosm experiment using seawater from Santa Barbara Channel amended with nutrients (Smith et al., 1995). Under simulated eutrophication conditions, the growth rates of bacteria from the Mediterranean sea ranged from 0.245 $h^{-1}$ to 0.853 $h^{-1}$ based on the data measured roughly every 24 hours in batch mesocosms (Lebaron et al., 1999). We would like to point out that our experiments were conducted in eutrophic coastal seawaters with reduced predatory grazing pressure due to seawater filtration, which could stimulate the net bacterial growth rate. In addition, some species belong to Pseudomonas group, one of the most abundant bacterioplankton group from outside, were reported to have high growth rates (Adav, 2008; Eagon, 1962). Therefore, we think choosing 0.5 $h^{-1}$ as the bacterial growth rate of the bacterioplankton is tenable. In our study, assuming the bacterioplankton concentration at day 2 representing the concentration of Pseudomonadaceae, one of the most abundant bacterioplankton groups from surrounding seawater, the concentration of Pseudomonadaceae at day 0 could be estimated based on the growth rate of 0.5 $h^{-1}$ and the bacterioplankton concentration ($6.693 \times 10^9$ cells/ml) at day 2. The estimated concentration of Pseudomonadaceae at day 0 was about 3 cells/ml. Therefore, the ratio of bacteria being continuously introduced to actual standing stocks in the mesocosms was low, which allowed us to detect potential $CO_2$ effects in this mesocosm experiment (Page 18 , Line 4-21).

**The derivation procedure is as follows**:
μ max ($h^{-1}$)=0.5
μ max ($h^{-1}$) = (Ln X- Ln Y)/48 hours
Y: the estimated concentration of Pseudomonadaceae at day 0 based on growth rate of *Pseudomonas aeruginosa* (μ ($h^{-1}$)=0.5).
X= $6.693 \times 10^9$ cells/ml (the average bacterioplankton concentration at day 2)
Y=X/$e^{48}$□

[revised manuscript text omitted]

---

## Author Response (AR5)

Dear Dr. Schulz:

We would like to thank you for your editorial handling of our paper (Interactive network configuration maintains bacterioplankton community structure under elevated $CO_2$ in a eutrophic coastal mesocosm experiment, bg-2017-10). In response to your comments, we have performed further analyses. Our responses are placed under your comments.

Associate Editor Decision: Publish subject to minor revisions (review by editor) (23 Nov 2017) by Kai G. Schulz
Comments to the Author:
Dear authors,
thank you for your reply. I am satisfied with your revision but may I encourage you to build your line of evidence in a slightly different manner. Instead of choosing a more or less arbitrary growth rate, I would suggest to calculate the minimum growth rate necessary for introducing, for instance, less than 1 permil of the standing stock. Then you could argue that this growth rate is reasonable in comparison to observed rates in oligotrophic communities, especially in the absence of significant grazing pressure.

Finally, could you please thoroughly check the manuscript for grammar.

Sincerely,
Kai Schulz

Response:
The average marine bacterial growth rates, mainly in oligotrophic seawaters, have been reported to be 1.1 day$^{-1}$ $\pm 0.83$ (Kirchman, 2016). However, the bacterial growth rates under eutrophic conditions are much higher than under oligotrophic conditions, and it has been reported to be on the order of per hour (Paul A. White, Jacob Kalff, 1991). The bacterial growth rate reached 16.2 day$^{-1}$ (0.675 h$^{-1}$) during a diatom bloom in a mesocosm experiment using seawater from Santa Barbara Channel amended with nutrients (Smith et al., 1995). Under simulated eutrophic conditions, the growth rates of bacteria from the Mediterranean Sea ranged from 0.245 h$^{-1}$ to 0.853 h$^{-1}$ based on the data measured roughly every 24 hours in batch mesocosms (Philippe and Al, 1999). Assuming the daily bacterioplankton introduction from outside was less than 0.1% of the standing stocks (the average bacterioplankton biomass on day 2, $6.693 \times 10^9$ cells/ml), the minimum bacterial growth rate can be calculated. The calculated minimum bacterial growth rate was 0.14 h$^{-1}$, which is reasonable in comparison to observed bacterial growth rates in eutrophic and oligotrophic communities. Furthermore, our experiments were conducted in eutrophic coastal seawaters with reduced predatory grazing pressure due to seawater filtration, which could stimulate the net bacterial growth rate. Therefore, the ratio of bacteria being continuously introduced to actual standing stocks in the mesocosms was low, allowing us to detect potential $CO_2$ effects in this mesocosm experiment (Page 18, Line 4-19).

**The derivation procedure is as follows**:

The bacterioplankton introduction from outside on day 0 and day 1:

$Y=(X*e^{24\mu}+X)*e^{24\mu}$

Y: $6.693 \times 10^9$ cells/ml (the average bacterioplankton abundance at day 2)

X: Daily bacterioplankton introduction from outside (assuming less than 0.1% of the standing stocks (less than $6.693 \times 10^6$ cells/ml)

$\mu=0.143749$ h$^{-1}$

The bacterioplankton introduction from outside only on day 0:

$Y=X*e^{48\mu}$

Y: $6.693 \times 10^9$ cells/ml (the average bacterioplankton abundance at day 2)

X: Daily bacterioplankton introduction from outside (assuming less than 0.1% of the standing stocks (less than $6.693 \times 10^6$ cells/ml)

$\mu=0.14375$ h$^{-1}$

References:

[revised manuscript text omitted]

a b

Alphaproteobacteria    Chloroplast    Gamaproteobacteria    Flavobacteriia

Betaproteobacteria    Saprospirae    Epsilonproteobacteria    Phycisphaerae

**Figure 4**

[Figure]

**Figure 5**

**Table 1** Topological properties of the bacterioplankton communities as represented by molecular networks under HC and LC treatments; also their rewired random networks.

| | Experimental network | | | | | | | Random network | | |
|---|---|---|---|---|---|---|---|---|---|---|
| | Total nodes | Total links | R2 of power-law | Average clustering coefficient (avgCC) | Average connectivity | Harmonic geodesic distance (HD) | Modularity | Average clustering coefficient (avgCC) | Harmonic geodesic distance (HD) | Modularity |
| LC | 85 | 209 | 0.817 | 0.402 | 0.625 | 3.397 | 0.414 | 0.424 +/- 0.023 | 2.187 +/- 0.049 | 0.249 +/- 0.010 |
| HC | 96 | 310 | 0.817 | 0.448 | 0.714 | 2.956 | 0.303 | 0.292 +/- 0.023 | 2.306 +/- 0.059 | 0.323 +/- 0.008 |

**Table 2** Dissimilarity tests of bacterial communities in the HC and LC treatments at various time points.

| | Anosim | | MRPP | | Adonis | |
|---|---|---|---|---|---|---|
| Time | R | P-value | $\delta$ | P-value | $R^2$ | P |
| day6 | -0.111 | 0.602 | 0.3952 | 1 | 0.15447 | 1 |
| day8 | 0.111 | 0.284 | 0.438 | 0.6 | 0.2 | 0.5 |
| day10 | 0.037 | 0.613 | 0.4929 | 0.7 | 0.17829 | 0.7 |
| day13 | 0.111 | 0.309 | 0.412 | 0.5 | 0.19714 | 0.5 |
| day19 | 0 | 0.693 | 0.4336 | 0.3 | 0.28263 | 0.3 |
| day29 | -0.259 | 1 | 0.4513 | 0.9 | 0.15517 | 0.9 |